# Adversarial Reweighting for Partial Domain Adaptation

**Xiang Gu, Xi Yu, Yan Yang, Jian Sun**,* and **Zongben Xu**
School of Mathematics and Statistics, Xi'an Jiaotong University, P.R. China
`{xianggu,ericayu,yangyan92}@stu.xjtu.edu.cn`
`{jiansun,zbxu}@xjtu.edu.cn`

## Abstract

Partial domain adaptation (PDA) has gained much attention due to its practical setting. The current PDA methods usually adapt the feature extractor by aligning the target and reweighted source domain distributions. In this paper, we experimentally find that the feature adaptation by the reweighted distribution alignment in some state-of-the-art PDA methods is not robust to the "noisy" weights of source domain data, leading to negative domain transfer on some challenging benchmarks. To tackle the challenge of negative domain transfer, we propose a novel Adversarial Reweighting (AR) approach that adversarially learns the weights of source domain data to align the source and target domain distributions, and the transferable deep recognition network is learned on the reweighted source domain data. Based on this idea, we propose a training algorithm that alternately updates the parameters of the network and optimizes the weights of source domain data. Extensive experiments show that our method achieves state-of-the-art results on the benchmarks of ImageNet-Caltech, Office-Home, VisDA-2017, and DomainNet. Ablation studies also confirm the effectiveness of our approach.

## 1 Introduction

Deep learning has achieved impressive success in image recognition [15, 20, 39]. However, deep learning models often rely on massive labeled training data, requiring an expensive and time-consuming labeling process. To alleviate the dependency of deep learning models on a large number of labeled data, domain adaptation (DA) [27] transfers the knowledge from a relevant source domain with rich labeled data to the target domain. The core of DA is to build a predictive model for the target domain using the training data of the source domain, and the model is expected to be robust to the distribution discrepancy (*a.k.a., domain shift*) between source and target domains. Domain adaptation methods often train the robust model by aligning distributions of different domains by moment matching [17, 19, 24, 28, 40, 48] or adversarial training [8, 25, 38, 40, 41, 42, 50]. The conventional closed-set DA methods generally assume that the source and target domains share the same label space. However, this assumption is often not realistic in practice. It is usually difficult to find a relevant source domain with identical label space as the target domain. This motivates the research on the learning problem of partial domain adaptation [3, 49].

Partial domain adaptation (PDA) is an important subcategory of domain adaptation. PDA [3, 4, 49] tackles the scenario that the label space of the target domain is a subset of that of the source domain. Partial domain adaptation is more challenging than vanilla closed-set domain adaptation. Because, besides the challenge of the domain shift, the existence of source-only classes that do not exist in the target domain can cause class-wise feature mismatch when aligning distributions. This potentially leads to *negative transfer* [27], *i.e.*, the DA approaches hurt the performance of learning in the target

---

*The corresponding author.

domain. To mitigate the negative transfer, current PDA methods [3, 4, 5, 21, 23, 34, 46, 49] commonly reweight the source domain data to decrease the importance of data belonging to the source-only classes. The target and reweighted source domain data are used to train the feature extractor by adversarial training [3, 4, 5, 23, 46, 49] or kernel mean matching [21, 34] to align distributions.

In this paper, we first propose to measure the hardness of a dataset for PDA using the probability of target domain data being misclassified into source-only classes. We then observe that some of the state-of-the-art reweighted distribution alignment losses, *e.g.*, PADA [4], BAA [23], and reweighted MMD [34], cause negative transfer on VisDA-2017 [32] and DomainNet [31] datasets that are more challenging than Office-31 [36], Office-Home [43] and ImageNet-Caltech [12, 35], according to our hardness measure. Specifically, learning/adapting the feature extractor by aligning the feature distributions of the reweighted source and target domain data can even worsen the performance of the baseline model without feature distribution alignment. We find that this negative domain transfer effect is mainly because that these reweighted feature distribution alignment methods are not robust to the "noises" of source data weights, *i.e.*, some source-only-class data are mistakenly assigned with non-zero weights in the alignment losses.

To tackle the negative domain transfer in PDA, we propose a novel adversarial reweighting (AR) approach, which adversarially learns to reweight the source domain data for aligning the distributions of the source and target domains. Specifically, our approach relies on adversarial training to learn the weights of source domain data to minimize the Wasserstein distance between the reweighted source domain and target domain distributions. The weight learning process is conducted in an adversarial reweighting model, using the dual form of the Wasserstein distance. We then define a reweighted cross-entropy loss on the reweighted source domain data and use the conditional entropy loss on the target data to train the transferable recognition network for the target domain. The network training and weight learning are performed alternately in an iterative training algorithm.

The current PDA methods [3, 4, 5, 21, 23, 34, 46, 49] design/learn source data weights based on the classifier [3, 4, 21, 23, 34, 46] or discriminator [5, 49]. They then train the feature extractor using a reweighted distribution alignment loss defined on the target and reweighted source data. Different from them, firstly, we learn the weights of source data in our proposed adversarial reweighting model to decrease the weight of source-only-class data. Secondly, we reduce the domain gap by reweighting the source domain data, instead of directly optimizing the feature extractor to match feature distributions This strategy may mitigate the negative domain transfer when the source domain data contain "noisy" weights. Note that Balaji *et al.* [2] and Huang *et al.* [16] also align data distributions by data reweighing. Huang *et al.* [16] proposes a shallow method that reweights the source data using the density ratio estimated by kernel mean matching for closed-set DA. Differently, we reweight the source data using an adversarial reweighting model for PDA in feature space in deep learning framework. Balaji *et al.* [2] proposes a robust optimal transport model to decrease the importance of the outliers of target data in distribution alignment for closed-set DA. Different from Balaji *et al.* [2], we reweight the source domain data to decrease the importance of source-only-class data in the source classification loss to mitigate the negative transfer for PDA.

We conduct extensive experiments on five benchmark datasets *i.e.*, Office-31, ImageNet-Caltech, Office-Home, VisDA-2017, and DomainNet. The results show that our method achieves state-of-the-art results for PDA on ImageNet-Caltech, Office-Home, VisDA-2017, and DomainNet datasets, and achieves competitive results on Office-31. Ablation studies indicate that our method significantly outperforms its baseline on all datasets. Our code is available at https://github.com/XJTU-XGU/Adversarial-Reweighting-for-Partial-Domain-Adaptation.

In the following sections, we discuss the limitations of feature adaptation by domain distribution alignment in Sect. 2. Section 3 presents the details of our approach. Section 4 reports the experimental results and Sect. 5 concludes this work.

## 2 Limitations of Feature Adaptation by Domain Distribution Alignment

In this section, we first introduce the setting of PDA and summarize the popular reweighted distribution alignment methods for PDA [3, 4, 5, 21, 23, 34, 49]. We then propose a hardness measure for PDA benchmarks and show the limitations of the reweighted distribution alignment.

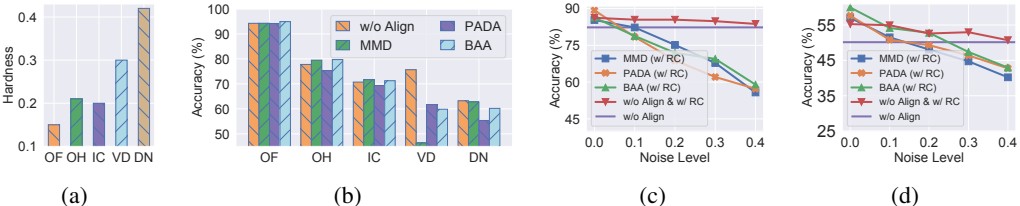

(a)          (b)          (c)          (d)

Figure 1: (a) Hardness of five benchmark datasets, *i.e.*, Office-31 (OF), Office-Home (OH), ImageNet-Caltech (IC), VisDA-2017 (VD), and DomainNet (DN), for PDA. (b) Comparison of results of different reweighted distribution alignment losses (PADA, BAA, and MMD) and the baseline (w/o Align) for PDA on five benchmark datasets. (c-d) Results for reweighted distribution alignment losses with simulated source data weights under varying noise levels in the tasks of (c) Synthetic (S) → Real (R) on VisDA-2017 and (d) Clipart (C) → Painting (P) on DomainNet.

**Problem setting.** In PDA, we are given a labeled dataset $\mathcal{S} = \{x_i^s, y_i^s\}_{i=1}^{n_s}$ from source domain and an unlabeled dataset $\mathcal{T} = \{x_j^t\}_{j=1}^{n_t}$ from target domain, where $x^s$ and $x^t \in \mathbb{R}^d$ respectively denote the source and target data, $y_i^s \in \mathcal{Y}^s$ is the label of $x_i^s$, and $\mathcal{Y}^s = \{0, 1, \cdots, K\}$ is the source label space. The goal is to train a recognition network to predict the label of $x_j^t$. Notably, in PDA, the target label space $\mathcal{Y}^t$ is a subset of $\mathcal{Y}^s$, *i.e.*, $\mathcal{Y}^t \subset \mathcal{Y}^s$, which is different from the setting ($\mathcal{Y}^t = \mathcal{Y}^s$) in the closed-set DA. The PDA methods often use a feature extractor $F$ to extract the features and a classifier $C$ to predict labels, and optionally use a discriminator $D$ to discriminate domains.

**Summary of reweighted distribution alignment methods in PDA.** The challenges of PDA come from two aspects: the domain shift between source and target domains, and the negative effect of source-only-class data in adaptation. To tackle these challenges, most popular PDA methods (*e.g.*, SAN [3], PADA [4], ETN [5], DRCN [21], BA³US [23], TSCDA [34], and IWAN [49]) commonly adapt the feature extractor by minimizing the feature distribution distance between the reweighted source domain and the target domain, measured by *the reweighted distribution alignment losses*. The weights in the reweighted distribution alignment losses are usually designed based on the output of the classifier or the discriminator. The widely adopted distribution distance metrics include the Maximization Mean Discrepancy (MMD) and the Jensen–Shannon (JS) divergence. Minimizing the MMD matches the kernel mean embedding of distributions in the Reproducing Kernel Hilbert Space. Minimizing the JS divergence is equivalent to the adversarial training as in the Generative Adversarial Network [10]. More comparisons of the PDA methods are given in Supp. A.

**Measuring the hardness of a dataset for PDA.** We propose to measure the hardness of a dataset for PDA using the probability of the target domain data being misclassified into the source-only classes. Specifically, we first train a model by minimizing the source classification loss and the entropy loss on target domain, which is the baseline for PDA. Then the hardness of a PDA task is defined as the average predicted probability of the target domain samples being misclassified into the source-only classes, using the trained model. The average hardness of all tasks in the dataset is taken as the hardness of the dataset. We report the hardness of five PDA benchmark datasets in Fig. 1(a). Figure 1(a) indicates that VisDA-2017 and DomainNet are more challenging than the other datasets.

**Adapting feature extractor by reweighted distribution alignment may lead to negative domain transfer.** We show the limitations of the reweighted distribution alignment in this paragraph. We report the average classification accuracies for different reweighted distribution alignment losses (including the reweighted adversarial training losses of PADA [4] and BAA [23], and the reweighted MMD loss in [34]) and the baseline (w/o Align) on five benchmark datasets in Fig. 1(b). We also give the detailed results in Supp. B. We can observe in Fig. 1(b) that additionally minimizing these reweighted distribution alignment losses by the feature extractor worsens the performance of the baseline on some challenging datasets, *e.g.*, VisDA-2017 and DomainNet. This indicates that adapting the feature extractor by the reweighted distribution alignment losses leads to negative transfer on these datasets. We next investigate the reasons for this finding.

**Adapting feature extractor by reweighted distribution alignment is not robust to noisy weights.** In the methods of [3, 4, 21, 23, 34], the weight of source domain data $(x_i^s, y_i^s)$ is defined as the average predicted probability for category of $y_i^s$ on the target domain data, *i.e.*, the $y_i^s$-th element

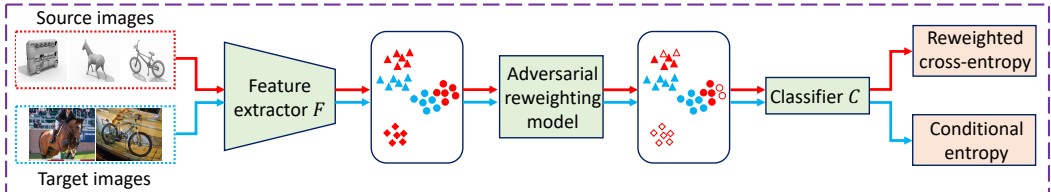

Figure 2: Architecture of our Adversarial Reweighting approach for PDA. Red (resp. blue) arrows indicate the computational flow for source (resp. target) domain data. Both source and target images are mapped to feature space by the feature extractor. Our adversarial reweighting model automatically reweights the importance of source domain data to match the target domain distribution in feature space to decrease the importance of the data of source-only classes. We define a reweighted cross-entropy loss on the reweighted source domain data distribution and the conditional entropy losses on target domain data to learn a transferable recognition network for the target domain.

of $\frac{1}{n_t} \sum_{j=1}^{n_t} C(F(x_j^t))$. When the domain gap between source and target domains is large, the prediction by the source classifier on the target domain may be uncertain. Then, the predicted probability of classifying the target data as source-only classes may be non-zero or even possibly significantly larger than zero. Hence, reweighting source classes based on the outputs of the classifier may assign non-zero weights to the source-only classes, *i.e.*, the designed weights may contain "noises". We analyze that the ineffectiveness of adapting the feature extractor by the reweighted distribution alignment could be because it is not robust to the "noisy" weights. To illustrate this, we conduct experiments with simulating the source domain data weights with different noise levels. We assign the noise to each source-only class in proportion to its weight predicted by the classifier aforementioned. If the weight predicted by the classifier is $c^i, i \in \mathcal{Y}^s$, for the noise level $p \in [0, 1]$, the simulated weight for each source-only class is $pc^i / \sum_{k \in \mathcal{Y}^s \setminus \mathcal{Y}^t} c^k$, for $i \in \mathcal{Y}^s \setminus \mathcal{Y}^t$. Similarly, the simulated weight for each source-shared class (the class exists in both source and target label spaces) is $(1-p)c^i / \sum_{k \in \mathcal{Y}^t} c^k$, for $i \in \mathcal{Y}^t$. If $p = 0$, the data of source-only classes are ideally assigned with weights of 0, and as the $p$ increases, these data are assigned with larger weights, which are taken as the "noisy" weights. We also use the simulated weights to reweight the importance of source data in the classification loss, denoted as "RC".

Figures 1(c) and 1(d) show the results for different reweighted distribution alignment losses with the simulated weights under varying noise levels. We can see that when the noise level is near zero, all the alignment losses outperform "w/o Align", indicating that the alignment losses lead to positive domain transfer in this case. However, as the noise level increases, the performance of the alignment losses decreases rapidly and even becomes significantly worse than that of "w/o Align". Specifically, in task S→R (Fig. 1(c)), when the noise level is 0.1, the performance of the alignment losses begin to be inferior to that of "w/o Align". Similarly, in task C→P (Fig. 1(d)), the performance of the alignment losses begin to be inferior to that of "w/o Align" at noise level 0.3. While the "real" noise level (the sum of weights based on the classifier over source-only classes) is larger than 0.3 (resp. 0.4) in the two tasks. Therefore, the negative domain transfer of the reweighted distribution alignment losses may be because of the "noise" in weights. It is also seen that when the noise level ranges from 0 to 0.4, the approach "w/o Align & w/ RC" that reweights data importance in source classification loss consistently outperforms the baseline "w/o Align" in both two tasks. This indicates that reweighting data importance in the source classification loss is more robust to weight noise, compared with the reweighted distribution alignment losses, in the two tasks.

The above observations indicate that adapting the feature extractor by the reweighted distribution alignment is not robust to the noise in source data weights, and can cause negative domain transfer on some challenging datasets. Surprisingly, reweighting data importance in the source classification loss may be more robust to weight noise than the reweighted distribution alignment losses.

## 3 Adversarial Reweighting for Partial Domain Adaptation

According to the observations in Sect. 2, adapting the feature extractor by the reweighted distribution alignment losses may cause negative domain transfer, when the domain gaps are large. In this work,

we propose a novel adversarial reweighting (AR) approach for PDA, which relies on adversarial learning of source data weights for aligning the distributions of source and target domain features. Figure 2 illustrates the idea of our approach. We map the source and target data to the feature space by the feature extractor $F$. We then design an adversarial reweighting model to reweight the importance of the source domain data in order to match the target domain distribution in the feature space. In this adversarial reweighting model, we reduce the domain gap by reweighting the source domain data features, instead of adapting the feature extractor. We define a reweighted cross-entropy loss on the source domain data (with the weights learned by the adversarial reweighting model) and a conditional entropy loss on the target domain data to learn a transferable recognition network for the target domain. In the following, we first introduce our network training loss, then discuss the detailed adversarial reweighting model, and finally present our training algorithm.

## 3.1 Loss for Training Recognition Network

The loss for training the recognition network of our approach is the combination of the reweighted cross-entropy loss on source domain and the conditional entropy loss on target domain, defined by

$$\mathcal{L}(\theta_F, \theta_C, \mathbf{w}) = \frac{1}{n_s} \sum_{i=1}^{n_s} w_i \mathcal{J}(C(F(x_i^s; \theta_F); \theta_C), y_i^s) + \frac{1}{n_t} \sum_{j=1}^{n_t} H(C(F(x_j^t; \theta_F); \theta_C)), \quad (1)$$

where $\mathcal{J}(\cdot, \cdot)$ is the cross-entropy loss defined by $\mathcal{J}(\mathbf{p}, y) = -\sum_k \mathbb{I}_{\{y=k\}} \log p_k$, $H(\cdot)$ is the conditional entropy defined as $H(\mathbf{p}) = -\sum_k p_k \log p_k$ for distribution $\mathbf{p} = (p_1, p_2, \cdots, p_K)^T$, $\mathbf{w} = (w_1, w_2, \cdots, w_{n_s})^T$ is the weights specifying the importance of source domain data such that $\sum_{i=1}^{n_s} w_i = n_s$, and $\theta_F$ and $\theta_C$ are respectively the parameters of $F$ and $C$. The weights are learned in an adversarial reweighting model in Sect. 3.2. Minimizing the reweighted cross-entropy loss enforces the recognition network to predict the labels of input images. Minimizing the conditional entropy loss encourages the low-density separation between classes [11] on target domain.

## 3.2 Adversarial Reweighting Model

Following [5], we assume that the source domain data of shared classes $\mathcal{Y}^t$ are closer to the target domain data than those source domain data belonging to the source-only classes $\mathcal{Y}^s \backslash \mathcal{Y}^t$. To decrease the importance of the source-only-class data and meanwhile reduce the domain shift, we learn the weights of source domain data by minimizing the Wasserstein distance between the reweighted source domain distribution and target domain distribution. The weight learning process is formulated as an adversarial reweighting model. We first introduce the Wasserstein distance.

**Wasserstein distance.** The Wasserstein distance is a metric that measures the discrepancy between two distributions. The Wasserstein distance between distributions $\mu$ and $\nu$ is defined by $W(\mu, \nu) = \min_{\pi \in \Pi} \mathbb{E}_{(x,y) \sim \pi} [\|x - y\|]$, where $\Pi$ is the set of couplings of $\mu$ and $\nu$, *i.e.*, $\Pi = \{\pi | \int \pi(x, y) dy = \mu(x), \int \pi(x, y) dx = \nu(y)\}$, and $\|\cdot\|$ is the $l_2$-norm. Leveraging the Kantorovich-Rubinstein duality, the Wasserstein distance has the dual form of $W(\mu, \nu) = \max_{\|f\|_L \leq 1} \mathbb{E}_{x \sim \mu}[f(x)] - \mathbb{E}_{x' \sim \nu}[f(x')]$, where the maximization is over all 1-Lipschitz functions $f : \mathbb{R}^d \to \mathbb{R}$. Following [14], for computing the Wasserstein distance, we parameterize $f$ by a neural network $D$ (discriminator) with parameters $\theta_D$. Then, the Wasserstein distance becomes

$$W(\mu, \nu) \approx \max_{\theta_D \in \Theta} \mathbb{E}_{x \sim \mu}[D(x; \theta_D)] - \mathbb{E}_{x' \sim \nu}[D(x'; \theta_D)], \quad (2)$$

where $\Theta = \{\theta_D : \|D(\cdot; \theta_D)\|_L \leq 1\}$. We enforce the constraint in Eq. (2) with the gradient penalty technique as in [14]. Equation (2) allows us to approximately compute the Wasserstein distance using gradient-based optimization algorithms on large-scale datasets. We give more details for computing the Wasserstein distance in Supp. C. Compared with the other popular statistical distances, *e.g.*, the JS divergence, the Wasserstein distance enjoys better continuity for learning distributions [1, 44].

**Adversarial reweighting.** Our adversarial reweighting (AR) model is defined in the feature space. We denote the extracted feature as $z_i^s = F(x_i^s; \theta_F)$ and $z_j^t = F(x_j^t; \theta_F)$ for source and target domain data. The empirical distribution of the target domain data $\mathcal{T}$ is denoted as $\mathcal{P}_\mathcal{T} = \frac{1}{n_t} \sum_{j=1}^{n_t} \delta(z_j^t)$, where $\delta(\cdot)$ is the Dirac delta function. The reweighted source domain distribution is denoted as $\mathcal{P}_\mathcal{S}(\mathbf{w}) = \frac{1}{n_s} \sum_{i=1}^{n_s} w_i \delta(z_i^s)$. We then automatically learn the weights in the following principled

model. Based on the aforementioned assumption that the source-only-class data are more distant from target domain data than the source data of shared classes, we minimize the Wasserstein distance between the reweighted source domain and target domain distributions to learn the weights as follows

$$\min_{\mathbf{w}\in\mathcal{W}} W(\mathcal{P}_\mathcal{S}(\mathbf{w}), \mathcal{P}_\mathcal{T}). \tag{3}$$

To avoid the mode collapse, *i.e.*, the reweighted distribution is only supported on few data, we enforce $\sum_{i=1}^{n_s}(w_i-1)^2 < \rho n_s$. Then, the solution space is $\mathcal{W} = \{\mathbf{w} : \mathbf{w} = (w_1, w_2, \cdots, w_{n_s})^T, w_i \geq 0, \sum_{i=1}^{n_s} w_i = n_s, \sum_{i=1}^{n_s}(w_i-1)^2 < \rho n_s\}$. With the approximation of the dual form in Eq. (2), Eq. (3) is transformed to the following adversarial reweighting model:

$$\min_{\mathbf{w}\in\mathcal{W}} \max_{\theta_D\in\Theta} \frac{1}{n_s}\sum_{i=1}^{n_s} w_i D(z_i^s; \theta_D) - \frac{1}{n_t}\sum_{j=1}^{n_t} D(z_j^t; \theta_D). \tag{4}$$

In Eq. (4), the discriminator is trained to maximize (resp. minimize) the average of its outputs on the source (resp. target) domain to discriminate the source and target domains. Adversarially, the source data weights are learned to minimize the reweighted average of the outputs of the discriminator on the source domain. As a result, the source data (closer to the target domain) with smaller discriminator outputs will be assigned with larger weights. Therefore, defining the reweighted cross-entropy loss on the reweighted source data distribution encourages the transferability of the trained recognition network for the target domain. We will discuss the adversarial training of Eq. (4) in Sect. 3.3.

**Implementation techniques.** To better bridge the domain gaps (experimentally justified in Sect. 4.2), first, we use the spherical logistic regression (SLR) layer [13] as the classifier $C$. Second, following [49], the entropy loss in Eq. (1) is only used to update the feature extractor $F$ instead of both $F$ and $C$. The SLR layer outputs the cosine similarity of the target features and source prototypes [13]. With lower entropy, the learned target features need to be close to the source features so that the classifier (composed of source prototypes) can recognize them more surely. Therefore, our adversarial reweighting and entropy minimization can complement each other to reduce the domain gaps.

**Automatically adjusting $\rho$.** The proper magnitude of $\rho$ is important to our method. If the ratio $(|\mathcal{Y}^t|/|\mathcal{Y}^s|)$ of the sizes of label spaces of target to source domains is large, $\rho$ needs to be small to force more source data to contribute to the reweighted cross-entropy loss, and vice versa. Since the target label space is unknown, we automatically adjust $\rho$ to enforce the computed Wasserstein distance (the loss value in Eq. (4) and is denoted as $A^\rho$) in a preset interval $[A^{low}, A^{up}]$. To do this, we choose an initial value $\rho^0$ of $\rho$ and a constant $c > 1$, and adjust $\rho$ as follows. If $A^\rho > A^{up}$ (resp. $A^\rho < A^{low}$), we adjust $\rho$ by $\rho = c\rho$ (resp. $\rho = \rho/c$) and resolve Eq. (4). The adjustment is performed till $A^{low} \leq A^\rho \leq A^{up}$. We set $A^{up} = 5.0, A^{low} = -5.0, c = 1.2$, and $\rho^0 = 5.0$. We show that the performance of our method is not sensitive to these hyper-parameters, in Sect 4.2.

### 3.3 Training Algorithm

To train the recognition network to minimize the loss in Eq. (1), we alternately optimize the network parameters $(\theta_F, \theta_C)$ and learn the weights $\mathbf{w}$ by fixing others as known. We initialize $\mathbf{w}$ by $w_i = 1$ for all $i$. Then, we alternately run the following two procedures when training the network.

**Updating $\theta_F$ and $\theta_C$ with fixed w.** Fixing $\mathbf{w}$, we update $\theta_F$ and $\theta_C$ to minimize the loss in Eq. (1) for $M$ steps, using the mini-batch stochastic gradient descent algorithm.

**Updating w with fixed $\theta_F$ and $\theta_C$.** Fixing $\theta_F$ and $\theta_C$, we extract the features for all training data on both source and target domains, and learn $\mathbf{w}$ in Eq. (4). Since Eq. (4) is a min-max optimization problem, we can alternately optimize the weights $\mathbf{w}$ and the parameters $\theta_D$ of the discriminator by fixing the other one as known. For reducing the computational cost, we only perform the alternate optimization once, which yields satisfactory performance in experiments. Therefore, we first fix $w_i = 1$ for all $i$ and optimize $\theta_D$ to maximize the objective function in Eq. (4) using the gradient penalty technique, as in [14]. Then, fixing the discriminator, we optimize $\mathbf{w}$ as follows. We denote $d_i = D(z_i^s; \theta_D)$ and $\mathbf{d} = (d_1, d_2, \cdots, d_{n_s})^T$. The optimization problem for $\mathbf{w}$ becomes

$$\min_{\mathbf{w}} \mathbf{d}^T\mathbf{w}, \quad s.t. \ w_i \geq 0, \sum_{i=1}^{n_s}(w_i-1)^2 \leq \rho n_s, \sum_{i=1}^{n_s} w_i = n_s. \tag{5}$$

Equation (5) is a second-order cone program. We use the CVXPY [7] package to solve Eq. (5). We also automatically adjust $\rho$ as in Sect. 3.2.

Table 1: Accuracy (%) on Office-Home for partial domain adaptation.

| Method | Ar→Cl | Ar→Pr | Ar→Rw | Cl→Ar | Cl→Pr | Cl→Rw | Pr→Ar | Pr→Cl | Pr→Rw | Rw→Ar | Rw→Cl | Rw→Pr | Avg |
|---|---|---|---|---|---|---|---|---|---|---|---|---|---|
| ResNet-50 [15] | 46.33 | 67.51 | 75.87 | 59.14 | 59.94 | 62.73 | 58.22 | 41.79 | 74.88 | 67.40 | 48.18 | 74.17 | 61.35 |
| ADDA [42] | 45.23 | 68.79 | 79.21 | 64.56 | 60.01 | 68.29 | 57.56 | 38.89 | 77.45 | 70.28 | 45.23 | 78.32 | 62.82 |
| CDAN+E [25] | 47.52 | 65.91 | 75.65 | 57.07 | 54.12 | 63.42 | 59.60 | 44.30 | 72.39 | 66.02 | 49.91 | 72.80 | 60.73 |
| IWAN [49] | 53.94 | 54.45 | 78.12 | 61.31 | 47.95 | 63.32 | 54.17 | 52.02 | 81.28 | 76.46 | 56.75 | 82.90 | 63.56 |
| SAN [3] | 44.42 | 68.68 | 74.60 | 67.49 | 64.99 | 77.80 | 59.78 | 44.72 | 80.07 | 72.18 | 50.21 | 78.66 | 65.30 |
| PADA [4] | 51.95 | 67.00 | 78.74 | 52.16 | 53.78 | 59.03 | 52.61 | 43.22 | 78.79 | 73.73 | 56.60 | 77.09 | 62.06 |
| ETN [5] | 59.24 | 77.03 | 79.54 | 62.92 | 65.73 | 75.01 | 68.29 | 55.37 | 84.37 | 75.72 | 57.66 | 84.54 | 70.45 |
| DRCN [21] | 54.00 | 76.40 | 83.00 | 62.10 | 64.50 | 71.00 | 70.80 | 49.80 | 80.50 | 77.50 | 59.10 | 79.90 | 69.00 |
| SAFN [45] | 58.93 | 76.25 | 81.42 | 70.43 | 72.97 | 77.78 | 72.36 | 55.34 | 80.40 | 75.81 | 60.42 | 79.92 | 71.83 |
| RTNet$_{adv}$ [6] | 63.20 | 80.10 | 80.70 | 66.70 | 69.30 | 77.20 | 71.60 | 53.90 | 84.60 | 77.40 | 57.90 | 85.50 | 72.30 |
| BA$^3$US [23] | 60.62 | 83.16 | 88.39 | 71.75 | 72.79 | 83.40 | 75.45 | 61.59 | 86.53 | 79.25 | 62.80 | 86.05 | 75.98 |
| DPDAN [46] | 59.40 | – | 79.04 | – | – | – | – | – | 81.79 | 76.77 | 58.67 | 82.18 | – |
| SHOT [22] | 64.80 | 85.20 | **92.70** | 76.30 | 77.60 | **88.80** | 79.70 | 64.30 | 89.50 | 80.60 | **66.40** | 85.80 | 79.30 |
| Cls+Ent (w/ linear) | 54.03 | 73.61 | 83.27 | 69.51 | 67.56 | 77.75 | 69.51 | 53.73 | 83.38 | 74.56 | 59.34 | 82.41 | 70.72 |
| AR (w/ linear) (ours) | 62.13 | 79.22 | 89.12 | 73.92 | 75.57 | 84.37 | 78.42 | 61.91 | 87.85 | 82.19 | 65.37 | 85.27 | 77.11 |
| Cls+Ent | 61.61 | 78.21 | 86.20 | 73.19 | 71.76 | 79.62 | 75.11 | 59.76 | 86.31 | 79.16 | 61.67 | 83.59 | 74.68 |
| AR (ours) | 67.40 | 85.32 | 90.00 | 77.32 | 70.59 | 85.15 | 78.97 | 64.78 | 89.51 | 80.44 | 66.21 | 86.44 | 78.29 |
| Cls+Ent+AUS | 63.34 | 81.12 | 86.14 | 74.01 | 76.53 | 79.79 | 77.69 | 62.57 | 86.42 | 78.33 | 62.69 | 84.38 | 76.08 |
| AR+AUS (ours) | **68.24** | 85.60 | 90.61 | 75.91 | **77.54** | 81.89 | **81.73** | **66.39** | 89.01 | **83.65** | 65.61 | **86.95** | 79.43 |
| Cls+Ent+LS | 62.99 | 83.59 | 87.30 | 74.20 | 73.05 | 81.67 | 79.25 | 63.46 | 87.85 | 78.97 | 64.54 | 84.76 | 76.80 |
| AR+LS (ours) | 65.67 | **87.36** | 89.62 | **79.25** | 75.01 | 86.97 | 80.81 | 65.79 | **90.61** | 80.81 | 65.25 | 86.12 | **79.44** |

We give the pseudo-code of the training algorithm in Supp. D. When the size of source domain dataset is larger than 100k, solving Eq. (5) for all source data at once is infeasible. In such case, we randomly sample a subset (with size $N$) of the source dataset to update their weights $w$, and then update $\theta_F$ and $\theta_C$ using them, and iterate these two above procedures.

# 4   Experiments

We conduct experiments on five benchmark datasets to evaluate our adversarial reweighting (AR) approach, and compare it with state-of-the-art PDA methods.

**Datasets.** *Office-31* dataset [36] contains 4,652 images of 31 categories, collected from three domains: Amazon (A), DSLR (D), and Webcam (W). Following [3], we select images from the 10 categories shared by Office-31 and Caltech-256 [12] to build new target domains. *ImageNet-Caltech* is built with ImageNet (I) [35] and Caltech-256 (C) [12], respectively including 1000 and 256 classes. We utilize the 84 shared classes to build the target domain. As most networks are pre-trained on the training set of ImageNet, we use images from ImageNet validation set to build the target domain for task C→I. *Office-Home* [43] consists of four domains: Artistic (Ar), Clip Art (Cl), Product (Pr), and Real-World (Rw), sharing 65 classes. We use images of the first 25 classes in alphabetical order as the target domain. *VisDA-2017* [32] is a large-scale challenging dataset, containing two domains: synthetic (S) and real (R), with 12 classes. We use the first 6 classes in alphabetical order as the target domain. *DomainNet* [31] is another large-scale challenging dataset, composed of six domains with 345 classes. Since the labels of some domains and classes are very noisy, we follow [37] to adopt four domains (Clipart (C), Painting (P), Real (R), and Sketch (S)) with 126 classes. We use the first 40 classes in alphabetical order to build the target domain. On these datasets, we set every domain as the source domain in turn and use each of the rest domain(s) to build the target domain.

**Implementation details.** We implement our method using Pytorch [30] on a Nvidia Tesla v100 GPU. For the feature extractor $F$, we use the ResNet-50 [15] pre-trained on ImageNet [35], which excludes the last fully-connected layer. For the discriminator $D$, we use the same architecture with [8] (three fully connected layers with 1024, 1024 and 1 nodes), excluding the last sigmoid function. We use the SGD algorithm with momentum 0.9 to update $\theta_F$ and $\theta_C$. The learning rate of $\theta_C$ is ten times that of $\theta_F$. $\theta_D$ is updated by the Adam [18] algorithm with learning rate 0.001. Following [8], we adjust the learning rate $\eta$ of $\theta_C$ by $\eta = \frac{0.01}{(1+10p)^{-0.75}}$, where $p$ is the training progress linearly changing from 0 to 1. We set the batchsize to 36, $M = 500$, and $N = 36M$. For more details, please see Supp. E.

Table 2: Accuracy (%) on DomainNet for partial domain adaptation.

| Method | C→P | C→R | C→S | P→C | P→R | P→S | R→C | R→P | R→S | S→C | S→P | S→R | Avg |
|---|---|---|---|---|---|---|---|---|---|---|---|---|---|
| ResNet-50 [15] | 41.21 | 60.01 | 42.13 | 54.52 | 70.80 | 48.32 | 63.1 | 58.63 | 50.26 | 45.43 | 39.3 | 49.75 | 51.96 |
| DANN [9] | 27.83 | 36.64 | 29.91 | 31.79 | 41.98 | 36.58 | 47.64 | 46.81 | 40.85 | 25.82 | 29.54 | 32.72 | 35.68 |
| CDAN+E [25] | 37.46 | 48.26 | 46.61 | 45.50 | 60.96 | 52.63 | 62.01 | 60.63 | 54.74 | 35.37 | 38.50 | 43.63 | 48.86 |
| SAN [3] | 34.35 | 51.62 | 46.23 | 57.13 | 70.21 | 58.25 | 69.61 | 67.49 | 67.88 | 41.69 | 41.15 | 48.44 | 54.50 |
| PADA [4] | 22.49 | 32.85 | 29.95 | 25.71 | 56.47 | 30.45 | 65.28 | 63.35 | 54.17 | 17.45 | 23.89 | 26.91 | 37.41 |
| BA$^3$US [23] | 42.87 | 54.72 | 53.79 | 64.03 | 76.39 | 64.69 | **79.99** | 74.31 | **74.02** | 50.36 | 42.69 | 49.65 | 60.63 |
| Cls+Ent (w/ linear) | 50.14 | 64.05 | **59.81** | 65.26 | 76.12 | 69.50 | 75.54 | 69.74 | 68.55 | 50.63 | 54.95 | 54.44 | 63.23 |
| AR (w/ linear) (ours) | **56.70** | **70.36** | 58.56 | 65.63 | 74.80 | **74.85** | 75.22 | 71.17 | 69.08 | **53.90** | **55.70** | **63.09** | **65.76** |
| Cls+Ent | 49.40 | 65.69 | 58.89 | 65.92 | 74.82 | 70.77 | 75.87 | 70.72 | 68.26 | 50.45 | **55.70** | 62.23 | 64.06 |
| AR (ours) | 52.66 | 68.24 | 58.29 | **66.78** | **77.53** | 74.38 | 76.70 | 71.77 | 70.48 | 53.66 | 53.60 | 61.57 | 65.47 |

Table 3: Accuracy (%) on Office-31, ImageNet-Caltech, and VisDA-2017 for PDA.

| Method | Office-31 | | | | | | | ImageNet-Caltech | | | VisDA-2017 | | |
|---|---|---|---|---|---|---|---|---|---|---|---|---|---|
| | A→D | A→W | D→A | D→W | W→A | W→D | Avg | C→I | I→C | Avg | R→S | S→R | Avg |
| ResNet-50 [15] | 83.44 | 75.59 | 83.92 | 96.27 | 84.97 | 98.09 | 87.05 | 71.29 | 69.69 | 70.49 | 64.28 | 45.26 | 54.77 |
| DAN [24] | 61.78 | 59.32 | 74.95 | 73.90 | 67.64 | 90.45 | 71.34 | 60.13 | 71.30 | 65.72 | 68.35 | 47.60 | 57.98 |
| DANN [9] | 81.53 | 73.56 | 82.78 | 96.27 | 86.12 | 98.73 | 86.50 | 67.71 | 70.80 | 69.23 | 73.84 | 51.01 | 62.43 |
| IWAN [49] | 90.45 | 89.15 | 95.62 | 99.32 | 94.26 | 99.36 | 94.69 | 73.33 | 78.06 | 75.70 | 71.30 | 48.60 | 59.95 |
| SAN [3] | 94.27 | 93.90 | 94.15 | 99.32 | 88.73 | 99.36 | 94.96 | 75.26 | 77.75 | 76.51 | 69.70 | 49.90 | 59.80 |
| PADA [4] | 82.17 | 86.54 | 92.69 | 99.32 | 95.41 | **100.0** | 92.69 | 70.48 | 75.03 | 72.76 | 76.50 | 53.50 | 65.00 |
| ETN [5] | 95.03 | 94.52 | 96.21 | **100.0** | 94.64 | **100.0** | 96.73 | 74.93 | 83.23 | 79.08 | – | – | – |
| DRCN [21] | 86.00 | 88.50 | 95.60 | **100.0** | 95.80 | **100.0** | 94.30 | 78.90 | 75.30 | 77.10 | 73.20 | 58.20 | 65.70 |
| RTNet$_{adv}$ [6] | 96.20 | 97.60 | 92.30 | **100.0** | 95.40 | **100.0** | 97.20 | – | – | – | – | – | – |
| BA$^3$US [23] | **99.36** | **98.98** | 94.82 | **100.0** | 94.99 | 98.73 | **97.81** | **83.35** | 84.00 | 83.68 | 67.56 | 69.86 | 68.71 |
| DPDAN [46] | 96.27 | 96.82 | **96.35** | **100.0** | 95.62 | **100.0** | 97.51 | – | – | – | – | 65.26 | – |
| Cls+Ent (w/ linear) | 90.45 | 87.80 | 94.68 | **100.0** | 94.36 | 98.09 | 94.23 | 77.74 | 77.82 | 77.78 | 69.00 | 82.32 | 75.66 |
| AR (w/ linear) (ours) | 91.72 | 97.63 | 95.62 | **100.0** | 95.30 | **100.0** | 96.71 | 81.78 | 85.83 | 83.81 | 74.82 | 85.30 | 80.09 |
| Cls+Ent | 80.89 | 87.12 | 94.05 | 94.58 | 93.95 | 99.36 | 91.66 | 79.60 | 82.59 | 81.10 | 66.63 | 84.72 | 75.68 |
| AR (ours) | 96.82 | 93.54 | 95.51 | **100.0** | **96.04** | 99.67 | 96.93 | 82.24 | **87.12** | **84.69** | **78.52** | **88.75** | **83.62** |

## 4.1 Results

The results on Office-Home and DomainNet are reported in Tables 1 and 2, respectively. The results on Office-31, ImageNet-Catech, and VisDA-2017 are reported in Table 3. Note the results of the compared methods on DomainNet in Table 2 are obtained by running their official codes on DomainNet dataset. "Cls+Ent" to denote the baseline approach that minimizes entropy loss on the target domain and source classification loss without using reweighting in Eq. (1). Since we used the spherical logistic regression (SLR) layer as the classifier, for completely fair comparisons, we also report the results of the versions of our AR and Cls+Ent with a linear layer as the classifier (denoted as AR (w/ linear) and Cls+Ent (w/ linear), respectively).

On all five datasets, our methods of AR and AR (w/ linear) significantly outperform their baselines of Cls+Ent and Cls+Ent (w/ linear), respectively. On Office-Home, we also combine AR with the adaptive uncertainty suppression (AUS) loss [23] and label smoothing (LS), which are used in state-of-the-art (SOTA) methods of BAUS [23] and SHOT [22]. Our methods of AR+AUS and AR+LS consistently improve the performance of their baselines, respectively. Notably, AR+LS achieves the SOTA result of 79.44%. On DomainNet, both AR and AR (w/ linear) significantly outperform the compared methods, and AR (w/ linear) achieves the SOTA result of 65.76%. On ImageNet-Caltech and VisDA-2017 datasets, AR achieves the SOTA results of 84.69% and 83.62%, respectively. Note that on VisDA-2017 and DomainNet datasets, our approach AR (w/ linear) outperforms the previous SOTA method BA$^3$US [23] with a large margin (by 11.38% and 5.13%, respectively).

In Table 3, on Office-31, our approach AR (w/ linear) achieves competitive results compared with other methods. We can see that the accuracies of RTNet$_{adv}$, BA$^3$US, and DPDAN on Office-31 are higher than 97%. This indicates that the prediction of the classifier on the target domain may be reliable and the source weights based on the classifier may contain less noise. In such a case,

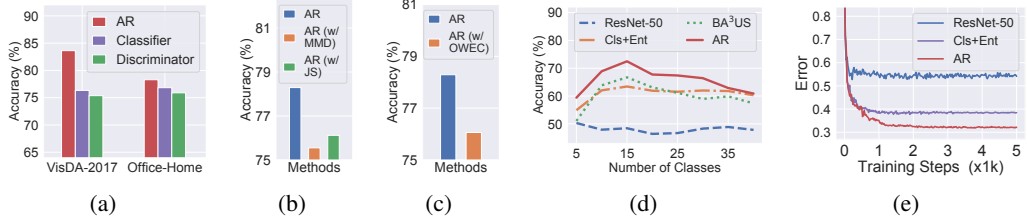

Figure 3: (a) Results for different reweighting strategies. (b) Ablation for MMD and JS-divergence for learning source data weights in our framework on Office-Home. (c) Ablation for learning one weight for each class (OWEC) on Office-Home. (d-e) Accuracy with varying number of target classes (d), convergence of test errors (e) in task Ar→Cl on Office-Home.

positive domain transfer may be achieved by feature adaptation. Note that although RTNet$_{adv}$ does not explicitly reweight data using the classifier outputs, the classifier outputs on the target domain are utilized as the component of state in a reinforcement learning framework for selecting the source data for feature adaptation. Hence, better classifier prediction may be beneficial to performance improvement. On the other four datasets, the average accuracies of compared methods are all lower than 85%, largely lower than the accuracies on Office-31. The prediction of the classifier on the target domain may not be reliable as Office-31, and our method outperforms the compared methods on these datasets. This may be because our proposed approach AR could be more robust to weight noise than other methods.

## 4.2 Analysis

**Comparison with the other reweighting strategies.** We compare different reweighting strategies for obtaining the weights in our loss of Eq. (1), including our adversarial reweighting (AR), reweighting based on the classifier as in [3, 4, 21, 23], and reweighting by the output of discriminator on source data as in [49]. The details of these strategies are given in Supp. A. The results in Fig. 3(a) show that our adversarial reweighting significantly outperforms the other two reweighting strategies on VisDA-2017 and Office-Home datasets, comfirming the effectiveness of our reweighting strategy.

**Ablation for MMD and JS-divergence to learn the weights.** We conduct ablation studies for JS-divergence and MMD to learn the weights in our framework (denoted as AR (w/ JS) and AR (w/ MMD), respectively), on Office-Home dataset. In Fig. 3(b), we can see that our AR using the Wasserstein distance outperforms AR (w/ JS) and AR (w/ MMD). When the supports of source and target distributions are disjoint, the Wasserstein distance may be more suitable to measure their distance than JS-divergence [1]. The MMD with widely used kernels may be unable to capture very complex distances in high dimensional spaces [1, 33], which may make it less effective than the Wasserstein distance in our framework.

**Ablation for learning one weight for each class (OWEC).** We conduct experiments for learning one weight for each class in our framework (denoted as AR (w/ OWEC)) on Office-Home dataset, as in Fig. 3(c). In Fig. 3(c), AR with learning the weight for each sample significantly outperforms AR (w/ OWEC) that learns one weight for each class. If the weight is learned for each sample, it is possible to assign higher weights to samples closer to the target domain, even in the same source class. The model trained in this case may be more transferable, because samples (even in the source-shared classes) less relevant to the target domain become less important.

**Accuracy with varying number of target classes.** We study our methods with different numbers of target classes in Fig. 3(d). Our method of AR significantly outperforms Cls+Ent and BA³US when the number of classes is smaller than 35. This shows that our method is effective for PDA when the label space mismatch between the source domain and the target domain is larger.

**Convergence.** In Fig. 3(e), we take the task Ar→Cl (Office-Home) as an example to study the convergence of our method. The approach of "ResNet-50" that trains the network using only source data converges fast but has high target test error. Our proposed AR achieves the lowest target error. We observe that our AR converges slightly slower than Cls+Ent, which may be because the source data weights in our reweighted cross-entropy loss are updated throughout the training process. We

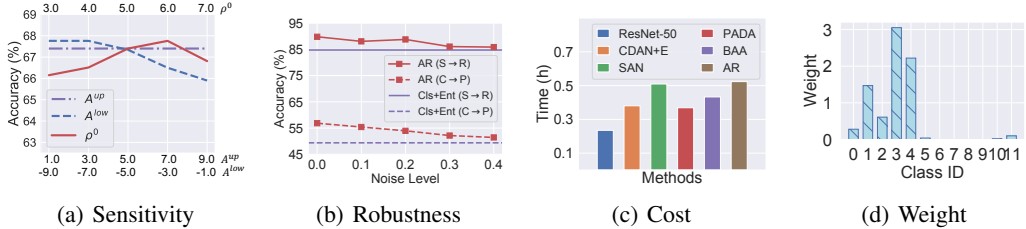

| (a) Sensitivity | (b) Robustness | (c) Cost | (d) Weight |

Figure 4: (a) Sensitivity to hyper-parameters in task Ar→Cl. (b) Robustness of AR to weight noise. (c) Computational cost of PDA methods. (d) Average weights for each class in task S→R.

report more empirical evidence, *e.g.*, the training loss, for justifying the convergence of the training algorithm in Supp. F.

**Sensitivity to hyper-parameters.** We testify the sensitivity of our method to hyper-parameters $A^{up}$, $A^{low}$, and $\rho^0$ mentioned in Sect. 3.2, as in Fig. 4(a). Figure 4(a) indicates that the performance of our proposed adversarial reweighting is not sensitive to these hyper-parameters.

**Robustness of AR to weight noise.** To verify the robustness of AR *w.r.t.* weight noise, we conduct simulation experiments for our method under different noise levels. The details of generating the simulated noisy weights are given in Supp. G, due to space limit. From Fig. 4(b), we observe that our AR consistently outperforms the baseline Cls+Ent in tasks C→P and S→R under different noise levels. This confirms the robustness of AR to weight noise.

**Computational cost.** We compare the computational cost of different methods with the total training time in the same training steps (5000 steps), as in Fig. 4(c). Figure 4(c) shows that our approach (AR) is comparable to other methods in terms of computational cost.

**Feature and weight visualization.** We visualize the learned average weights of source domain data for each class in task S→R, in Fig. 4(d). We can see that the source-shared-class (the first six classes) data get larger weights in general (except the sixth class). We also show the t-SNE embeddings [26] of learned features in task Ar→Cl (Office-Home) in Fig. 5. With the conditional entropy minimization, the target features in Fig. 5(a) learned by Cls+Ent are aligned with source features of all classes (including the source-only classes), which is unexpected in PDA. In Fig. 5(b), our proposed AR aligns the target features with partial source features. These visualizations may partially

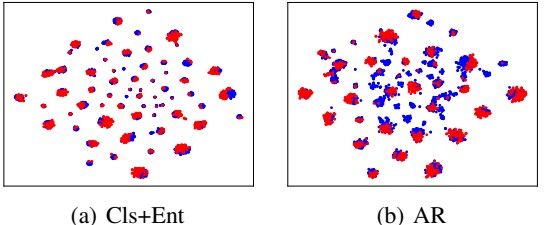

| (a) Cls+Ent | (b) AR |

Figure 5: (a-b) The t-SNE visualization of the learned source (blue) and target features (red) by Cls+Ent (a), and AR (b), in task Ar→Cl.

explain the success of our approach for tackling the problem of PDA. We also visualize more learned features and weights in Supp. H.

## 5   Conclusion

In this paper, we experimentally observe that adapting the feature extractor by reweighted distribution alignment is not robust to the "noisy" weights of source domain data and may hurt the performance of learning in target domain. We further propose a novel adversarial reweighting approach to tackle the problem of PDA. Extensive experiments show that our proposed approach achieves SOTA results on challenging datasets of ImageNet-Caltech, Office-Home, VisDA-2017, and DomainNet. In our future work, we are interested in applying our approach to the open-set DA [29] and universal DA [47].

## Acknowledgements

This work was supported by NSFC (11690011, U20B2075, 12090021, 11971373, U1811461, 61721002, 12026605) and National Key R&D Program 2018AAA0102201.

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
