# Adversarial Reweighting for Partial Domain Adaptation
## Supplementary Material

**Xiang Gu, Xi Yu, Yan Yang, Jian Sun, and Zongben Xu**
School of Mathematics and Statistics, Xi'an Jiaotong University, P.R. China
{xianggu,ericayu,yangyan92}@stu.xjtu.edu.cn
{jiansun,zbxu}@xjtu.edu.cn

## Supp. A  Methodology Comparisons of Feature-Adaptation-Based PDA Methods.

In this section, we compare the methodologies of the several feature-adaptation-based partial domain adaptation methods, as mentioned in Sect. 2. The losses of some of state-of-the-art PDA methods (*e.g.*, SAN [1], PADA [2], ETN [3], DRCN [9], BA$^3$US [10], TSCDA [12] and IWAN [15]) contain three terms, including the (reweighted) cross-entropy loss on source domain data, the conditional loss on target domain data, and the reweighted distribution alignment loss. The PDA methods [1, 2, 3, 9, 10, 12, 15] generally use a feature extractor $F$ to extract the features of data and a classifier $C$ to predict class labels. The general loss can be written as

$$\min_{F,C} \frac{1}{\sum_{l=1}^{n_s} w_l} \sum_{i=1}^{n_s} w_i \mathcal{J}(C(F(x_i^s)), y_i^s) + \text{Dist}(\mathcal{S}, \mathcal{T}; \mathbf{w}) + \frac{1}{n_t} \sum_{j=1}^{n_t} H(C(F(x_j^t))), \qquad \text{(S-1)}$$

where $\mathcal{J}(\cdot, \cdot)$ is the cross-entropy, $H(\cdot)$ is the conditional entropy, $\text{Dist}(\mathcal{P}_\mathcal{S}, \mathcal{P}_\mathcal{T}; \mathbf{w})$ is the reweighted distribution alignment loss for the target domain data and the reweighted source domain data, designed based on a distribution distance metric. $\mathbf{w} = (w_1, w_2, \cdots, w_{n_s})^T$ denotes the vector of the source domain data weights. These PDA methods are different in the following four aspects, including whether to reweight the importance of source data in the cross-entropy loss, the metric of adopted distribution distance, the strategy to obtain the source data weights, and whether to utilize the conditional entropy loss. The comparisons of the typical PDA methods are given in Table S-1.

Table S-1: Comparisons of losses of feature-adaptation-based partial domain adaptation methods.

| Method | Reweighting in CE | Reweighting Strategy | Distance Metric | Conditional Entropy |
|---|---|---|---|---|
| SAN [1] | ✗ | Classifier | JS | ✓ |
| IWAN [15] | ✗ | Discriminator | JS | ✓ |
| PADA [2] | ✓ | Classifier | JS | ✗ |
| ETN [3] | ✓ | Discriminator | JS | ✓ |
| TSCDA [12] | ✗ | Classifier | MMD | ✗ |
| DRCN [9] | ✗ | Classifier | MMD | ✗ |
| BA$^3$US [10] | ✓ | Classifier | JS | ✓ |

**Distribution distance metric.** The widely used distribution distances include the Maximization Mean Discrepancy (MMD) and the Jensen–Shannon (JS) divergence. The MMD is defined as

$$MMD(\mu, \nu) = \|\mathbb{E}_{x \sim \mu}[\phi(x)] - \mathbb{E}_{x' \sim \nu}[\phi(x')]\|_{\mathcal{H}}, \qquad \text{(S-2)}$$

where $\mathcal{H}$ is the Reproducing Kernel Hilbert Space and $\phi(\cdot) : \mathbb{R}^d \to \mathcal{H}$ is the feature map. Minimizing the MMD is to match the kernel mean embedding of distributions in the Reproducing Kernel Hilbert

35th Conference on Neural Information Processing Systems (NeurIPS 2021).

Space. The JS divergence satisfies

$$JS(\mu,\nu) \propto \max_{D} \mathbb{E}_{x\sim\mu}\left[\log D(x)\right] + \mathbb{E}_{x'\sim\nu}\left[\log(1 - D(x'))\right]. \tag{S-3}$$

Minimizing the JS divergence by $F$ is equivalent to the adversarial training on $F$ and $D$ as in the Generative Adversarial Network [5].

**Strategies for designing the source data weights.** The source data weights are designed based on the output of classifier $C$ [1, 2, 9, 10, 12, 14] or discriminator $D$ [3, 15]. If the strategy is based on the classifier, the weight of source data $(x_i^s, y_i^s)$ is defined by the average predicted probability for the category of $y_i^s$ on the target domain data, *i.e.*,

$$w_i \propto \mathbf{p}_{y_i^s}, \quad \mathbf{p} = \frac{1}{n_t}\sum_{j=1}^{n_t} C(F(x_j^t)), \tag{S-4}$$

where $\mathbf{p}_{y_i^s}$ is the $y_i^s$-th element of vector $\mathbf{p}$. If the strategy is based on the discriminator $D$ that is trained to predict the source (resp. target) domain label 1 (resp. 0), the weight of $(x_i^s, y_i^s)$ is defined by the predicted probability of $x_i^s$ belonging to the target domain by

$$w_i \propto 1 - D(F(x_i^s)). \tag{S-5}$$

## Supp. B    Detailed Results for Fig. 1(b)

We have reported the average classification accuracy over all tasks on each dataset in Fig. 1(b) of the paper. We now additionally report the detailed classification accuracies for all tasks on each dataset in this section. The detailed classification accuracies on Office-Home dataset and DomainNet dataset are shown in Tables S-2 and S-3, respectively. The detailed classification accuracies on Office-31, ImageNet-Caltech, and VisDA-2017 are shown in Table S-4.

Table S-2: Accuracy (%) comparisons for different reweighted distribution alignment losses and the baseline (w/o Align) for PDA on Office-Home dataset.

| Alignment loss | Ar→Cl | Ar→Pr | Ar→Rw | Cl→Ar | Cl→Pr | Cl→Rw | Pr→Ar | Pr→Cl | Pr→Rw | Rw→Ar | Rw→Cl | Rw→Pr | Avg |
|---|---|---|---|---|---|---|---|---|---|---|---|---|---|
| w/o Align | 54.03 | **73.61** | 83.27 | 69.51 | **67.56** | 77.75 | 69.51 | 53.73 | 83.38 | 74.56 | 59.34 | 82.41 | 70.72 |
| MMD | **57.31** | 73.11 | **84.59** | 69.70 | 67.17 | 78.63 | **69.79** | **56.36** | **85.15** | 75.11 | 61.61 | 81.79 | **71.69** |
| PADA | 55.94 | 68.52 | 82.27 | 68.23 | 64.03 | 76.15 | 67.68 | 52.84 | 83.38 | 74.38 | 57.31 | 81.12 | 69.32 |
| BAA | 54.57 | 73.17 | 84.48 | **69.79** | 67.39 | **79.85** | 69.24 | 53.25 | 84.59 | **76.31** | **59.82** | **82.69** | 71.26 |

Table S-3: Accuracy (%) comparisons for different reweighted distribution alignment losses and the baseline (w/o Align) for PDA on DomainNet dataset.

| Method | C→P | C→R | C→S | P→C | P→R | P→S | R→C | R→P | R→S | S→C | S→P | S→R | Avg |
|---|---|---|---|---|---|---|---|---|---|---|---|---|---|
| w/o Align | **50.14** | **64.05** | **59.81** | **65.26** | **76.12** | 69.5 | 75.54 | 69.74 | 68.55 | **50.63** | **54.95** | 54.44 | **63.23** |
| MMD | 49.95 | 59.18 | 56.71 | 65.11 | 75.2 | **70.33** | **76.01** | **71.16** | **70.14** | 50.28 | 52.17 | **57.2** | 62.79 |
| PADA | 39.43 | 56.49 | 48.89 | 54.21 | 68.41 | 63.19 | 72.71 | 68.27 | 64.77 | 39.63 | 41.36 | 46.93 | 55.36 |
| BAA | 42.93 | 57.64 | 53.75 | 59.62 | 73.45 | 68.51 | 75.12 | 71.11 | 67.16 | 46.04 | 46.64 | 60.39 | 60.20 |

Table S-4: Accuracy (%) comparisons for different reweighted distribution alignment losses and the baseline (w/o Align) for PDA on Office-31, ImageNet-Caltech, and VisDA-2017.

| Method | Office-31 | | | | | | | ImageNet-Caltech | | | VisDA-2017 | | |
|---|---|---|---|---|---|---|---|---|---|---|---|---|---|
| | A→D | A→W | D→A | D→W | W→A | W→D | Avg | C→I | I→C | Avg | R→S | S→R | Avg |
| w/o Align | 90.45 | 87.8 | 94.68 | **100.0** | 94.36 | **98.09** | 94.23 | 77.74 | 77.82 | 77.78 | 69.00 | **82.32** | **75.66** |
| MMD | 90.45 | 88.81 | 93.95 | **100.0** | 94.26 | **98.09** | 94.26 | **77.93** | 81.06 | 79.50 | 54.73 | 38.00 | 46.37 |
| PADA | 90.45 | 87.8 | 94.36 | **100.0** | 94.15 | **98.09** | 94.14 | 75.26 | 75.37 | 75.32 | 65.93 | 57.44 | 61.69 |
| BAA | **91.72** | **89.15** | **95.51** | **100.0** | **94.99** | **98.09** | **94.91** | 77.6 | **81.83** | **79.72** | **71.33** | 48.38 | 59.86 |

## Supp. C    Details for Computing the Wasserstein Distance

This section illustrates the details for computing the Wasserstein distance discussed in Sect. 3.2 of the paper. To compute the following approximation of the dual form of Wasserstein distance

$$W(\mu, \nu) \approx \max_{\{\theta_D : \|D(\cdot; \theta_D)\|_L \leq 1\}} \mathbb{E}_{x \sim \mu}[D(x; \theta_D)] - \mathbb{E}_{x' \sim \nu}[D(x'; \theta_D)], \tag{S-6}$$

we utilize the gradient penalty technique as in [6] to enforce the constraint in Eq. (S-6). Hence,

$$\theta_D^* = \arg \max_{\theta_D} \mathbb{E}_{x \sim \mu}[D(x; \theta_D)] - \mathbb{E}_{x' \sim \nu}[D(x'; \theta_D)] - \mathbb{E}_{\tilde{x} \sim \tilde{P}(\mu, \nu)}[(\|\nabla_{\tilde{x}} D(\tilde{x})\| - 1)^2], \tag{S-7}$$

where $\tilde{P}(\mu, \nu)$ denotes the samples uniformly along lines between pairs of points sampled from distributions $\mu$ and $\nu$. Equation (S-7) can be solved using the gradient-based optimization algorithm, *e.g.*, Adam algorithm [8]. With Eq. (S-7), the Wasserstein distance can be approximated by

$$W(\mu, \nu) \approx \mathbb{E}_{x \sim \mu}[D(x; \theta_D^*)] - \mathbb{E}_{x' \sim \nu}[D(x'; \theta_D^*)]. \tag{S-8}$$

## Supp. D    Pseudo-Code of Training Algorithm

Algorithm 1 presents the pseudo-code of our training algorithm in Sect. 3.3 of the paper.

---
**Algorithm 1** Training Algorithm.

---
**Input:** Datasets $\mathcal{S}$ and $\mathcal{T}$; Total training steps $N_{steps}$
**Output:** Trained network parameters $\theta_F, \theta_C$

1:  $\mathcal{S}' = \mathcal{S}$
2:  Initialize $\mathbf{w}$ by $w_i = 1, i = 1, 2, \cdots, |S'|$
3:  **for** $k \in \{0, 1, \cdots, N_{steps}\}$ **do**
4:      Sample a mini-batch data $(X_s, Y_s)$ and $X_t$ from $\mathcal{S}'$ and $\mathcal{T}$, respectively
5:      Update $\theta_F$ and $\theta_C$ with the loss in Eq. (1) computed on $(X_s, Y_s)$ and $X_t$, using the SGD
6:      **if** $k \% M = 0$ and $k > 0$ **then**
7:          **if** $|\mathcal{S}| > 10^5$ **then**
8:              Randomly select $N$ samples from $\mathcal{S}$ to construct $\mathcal{S}'$
9:          **end if**
10:         Extract features for all training data in both $\mathcal{S}'$ and $\mathcal{T}$
11:         Train $\theta_D$ as in Eq. (S-7) with $\mu = \mathcal{P}_{\mathcal{S}'}$ and $\nu = \mathcal{P}_{\mathcal{T}}$
12:         **while** True **do**
13:             Set $\rho = \rho^0$
14:             Solve Eq. (5) to update $\mathbf{w}$ and compute $A^\rho$
15:             **if** $A^\rho > A^{up}$ **then**
16:                 Set $\rho = c\rho$
17:             **else**
18:                 **if** $A^\rho < A^{low}$ **then**
19:                     Set $\rho = \rho / c$
20:                 **else**
21:                     Break
22:                 **end if**
23:             **end if**
24:         **end while**
25:     **end if**
26: **end for**

---

## Supp. E    Full Implementation Details.

The full implementation details are provided in this section. We implement our method using Pytorch [11] on a Nvidia Tesla v100 GPU. For the feature extractor $F$, we use the ResNet-50 [7] pre-trained on ImageNet [13], which excludes the last fully-connected layer. For the discriminator $D$, we use the same architecture with [4] (three fully connected layers with 1024, 1024 and 1 nodes),

excluding the last sigmoid function. We use the SGD algorithm with momentum 0.9 to update $\theta_F$ and $\theta_C$. The learning rate of $\theta_C$ is ten times that of $\theta_F$. $\theta_D$ is updated by the Adam [8] algorithm with learning rate 0.001. Following [4], we adjust the learning rate $\eta$ of $\theta_C$ by $\eta = \frac{0.01}{(1+10p)^{-0.75}}$, where $p$ is the training progress linearly changing from 0 to 1. We set the batchsize to 36, $M = 500$, and $N = 36M$. In the adaptation tasks with large-scale source (*e.g.*, task S→R on VisDA-2017 dataset, and all the tasks on ImageNet-Caltech and DomainNet datasets), we sample the mini-batch source data according to the learned weights (using a weighted random sampler) to calculate the classification loss. We find this strategy is more stable than the strategy that reweights the classification loss for each data in the uniformly sampled mini-batch, on these tasks.

## Supp. F    Justifications on the Convergence of Training Algorithm

This section provides more evidence for justifying the convergence of the training algorithm, which is the supplementary material to the discussion of convergence in Sect. 4.2 of the paper. We plot the training losses in Eq. (1), the approximated distance in Eq. (2), and the relative difference of weights in Fig. 1. The relative difference is $\frac{\|\Delta\mathbf{w}^t\|}{\|\mathbf{w}^t\|}$, where $\Delta\mathbf{w}^t = \mathbf{w}^{t+1} - \mathbf{w}^t$, and $\mathbf{w}^t$ is the value of the weights in the $t$-th iteration of the alternate training algorithm.

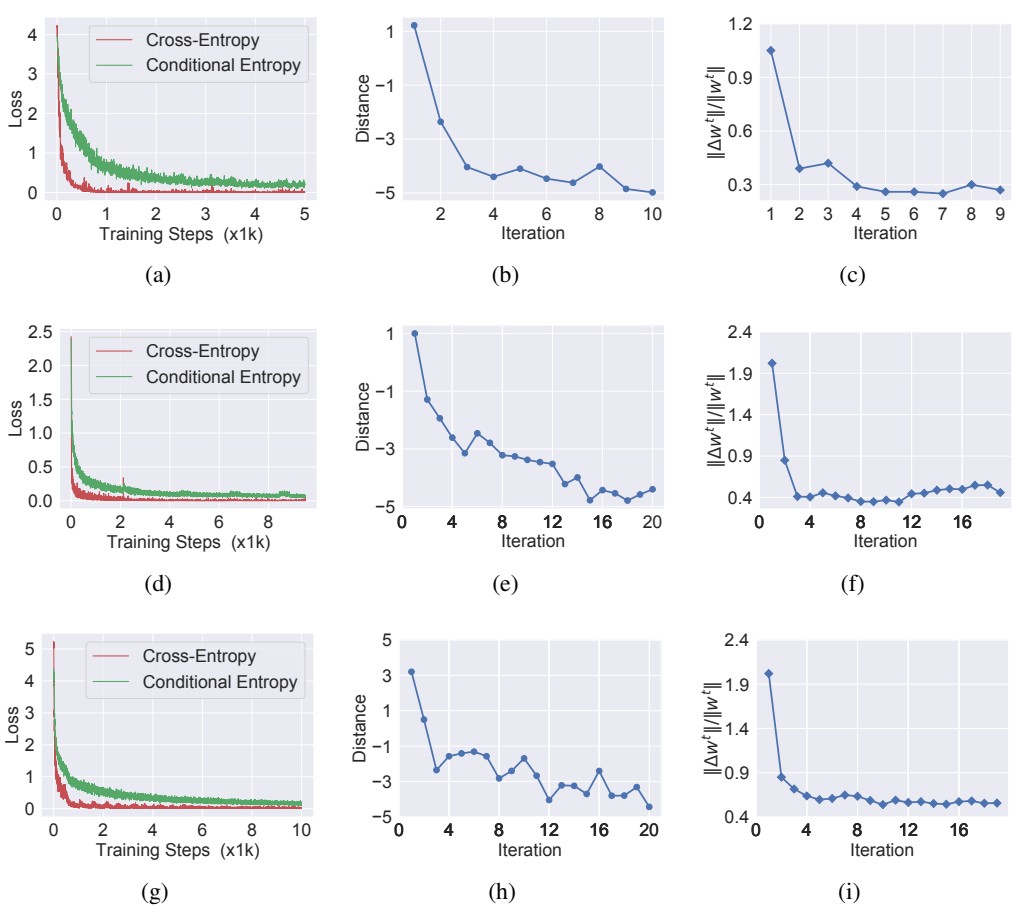

Figure 1: The training losses (a, d, and g), the approximated distance (b, e, and h), and the relative difference of weights (c, f, and i) in tasks of Ar→Cl on Office-Home (a-c), S→R on VisDA-2017 (d-f) and C→P on DomainNet (g-i).

In Figs. 1(a), 1(d), and 1(g), the training losses of our approach decrease stably and converge as the training processes. The fluctuation of the approximated distances in Figs. 1(b), 1(e), and 1(h),

and the non-zero value of $\frac{\|\Delta \mathbf{w}^t\|}{\|\mathbf{w}^t\|}$ in Figs. 1(c), 1(f), and 1(i) may be mainly because that the feature extractor is optimized by the SGD with approximated gradients over mini-batch.

## Supp. G    Details for Verifying the Robustness of AR to Weight Noise

To verify the robustness of AR *w.r.t.* weight noise, we conduct simulation experiments for our method under different noise levels. Specifically, we first obtain the source data weights $w_i, i = 1, \cdots, n_s$, through our adversarial reweighting model. For the noise level $p \in [0, 1]$, we simulate the weight of the $i$-th sample that belongs to source-only classes by $pw_i / \sum_j I(y_j \in \mathcal{Y}^s \backslash \mathcal{Y}^t) w_j$, for $i$ such that $i \in \mathcal{Y}^s \backslash \mathcal{Y}^t$ (*i.e.*, the $i$-th source sample that belongs to source-only classes is assigned with noisy weight in proportional to the weight learned by our adversarial reweighting model). Similarly, the $i$-th sample that belongs to source-shared classes is assigned noisy weight by $(1 - p)w_i / \sum_j I(y_j \in \mathcal{Y}^t) w_j$, for $i$ such that $i \in \mathcal{Y}^t$. The results are reported in Fig. 4(b) in the paper.

## Supp. H    Visualization of Learned Weights

We visualize the learned weights of source domain data in Fig. 2. Form Fig. 2, we can observe that the source data distant from target domain get smaller weights.

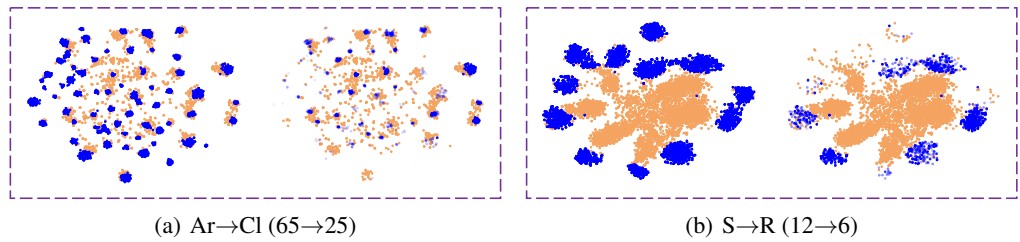

(a) Ar→Cl (65→25)         (b) S→R (12→6)

Figure 2: The learned weights of source data in tasks of Ar→Cl (a) and S→R (b). In both Figs. (a) and (b), the left figure plots the learned source (blue) and target (brown) features. The right figure plots the weighted source (blue) features and target (brown) features, where the source data points with more clear blue color are with larger weights.