# OpenReview forum: "Adversarial Reweighting for Partial Domain Adaptation"
_NeurIPS.cc/2021/Conference — NeurIPS 2021 Poster_

### Official Review · Reviewer_Xqv1 · 2021-07-16

**Rating:** 5
**Confidence:** 5

**Summary:**

This paper proposes a new method called Adversarial Reweighting (AR) for partial domain adaptation (PDA) where the target label space is a subset of the source label space. AR aims to estimate the weight of source instances via an adversarial training manner and Wasserstein distance and then exploits the weight to train a simple objective (re-weighted cross-entropy on source + entropy minimization on target). Results on several PDA benchmarks verify the effectiveness of AR.

**Limitations And Societal Impact:**

See my comments above.

**Main Review:**

Pros:
Different from previous PDA methods relying on class-level weights, the proposed AR method explores Wasserstein distance and learns the instance-level weight for each target sample, which is interesting and new in this field.
Promising results on several PDA benchmarks, especially 78.3 on the Office-Home dataset.

Cons:
the results of [BA^3US, ECCV2020] are not fairly compared since a weak variant BAA is compared throughout this paper. It seems that BA^3US performs better for Office and some other tasks like I->C.
it is also very confusing that the baseline method shown in each table achieves much higher results than previous PDA methods.
In figure3, the parameters sound sensitive, how do the authors determine these parameters for an unsupervised domain adaptation task? In fig.3(b), the authors need to compare AR with other PDA methods like BA^3US to study the effectiveness of AR.

minor comments:
if the Wasserstein distance is replaced by a vanillna JS distance, how AR performs for these PDA tasks?

[BA^3US, ECCV2020] Jian Liang, Yunbo Wang, Dapeng Hu, Ran He, and Jiashi Feng. A balanced and uncertainty-aware approach for partial domain adaptation. In ECCV, 2020.

**Time Spent Reviewing:**

6 hours

---

> ### Author Response · Authors · 2021-08-10
> **Response to Reviewer Xqv1 Part (2/2)**
>
>
> **Q3**: Determining the parameters.
>
> Since there is no ground truth label on the target domain, it's difficult to select proper hyper-parameters in DA. For example, it is impossible to do cross-valiation on target domain. We determine the hyper-parameters in our method with reference to [R1].  Figure 3(d) in the main paper shows that our selected parameters do not achieve the best result. There are a few works study model selection for DA [R1, R2]. We are interested in investigating this problem in our future work.
>
> **Q4**: Comparison with BA$^3$US with varying number of target classes.
>
> We compare our AR with BA$^3$US with varying number of target classes in the task Ar$\rightarrow$Cl on Office-Home dataset, in Table r3-6.
>
> Table r3-6. Results of AR and BA$^3$US with varying number of target classes in the task Ar$\rightarrow$Cl.
>
> |Number of classes|5|10|15|20|25|30|35|40|
> |:--|:--:|:--:|:--:|:--:|:--:|:--:|:--:|:--:|
> |BA$^3$US|51.04|63.85|66.72|63.12|61.14|58.96|59.89|57.37|
> |AR (ours)|__59.34__|__68.86__|__72.75__|__68.03__|__67.40__|__66.78__|__63.35__|__61.63__|
>
> From Table r3-6, varying number of classes from 0 to 40, the results show that our AR consistently outperforms BA$^3$US in the task Ar$\rightarrow$Cl. The results will be included in Fig. 3(b).
>
>
> **Q5**: Ablation on JS-divergence in our framework.
>
> We conduct an ablation study for JS-divergence to learn the weights in our framework (denoted as AR (w/ JS)), on Office-Home dataset. The results are reported in Table r3-7.
>
> Table r3-7. Ablation study for JS-divergence.
>
> |Method|Ar$\rightarrow$Cl|Ar$\rightarrow$Pr|Ar$\rightarrow$Rw|Cl$\rightarrow$Ar|Cl$\rightarrow$Pr|Cl$\rightarrow$Rw|Pr$\rightarrow$Ar|Pr$\rightarrow$Cl|Pr$\rightarrow$Rw|Rw$\rightarrow$Ar|Rw$\rightarrow$Cl|Rw$\rightarrow$Pr|Avg|
> |:---|:---:|:---:|:---:|:---:|:---:|:---:|:---:|:---:|:---:|:---:|:---:|:---:|:---:|
> |AR (w/ JS)| 61.37|80.50|89.73|74.12|73.83|80.89|76.58|61.31|86.69|__80.53__|63.28|84.65|76.12
> |AR|__67.40__|__85.32__|__90.00__|__77.32__|70.59|__85.15__|__78.97__|__64.78__|__89.51__|80.44|__66.21__|__86.44__|__78.29__|
>
>
> We can see that our AR using the Wasserstein distance outperforms AR (w/ JS). When the supports of source and target data distributions are disjoint, the Wasserstein distance may be more suitable to measure their distance than JS-divergence [1]. This may account for the improvement of performance achieved by AR over AR (w/ JS).The results will be included in the main paper.
>
>
>
> [R1] Sugiyama, et al., Covariate Shift Adaptation by Importance Weighted Cross Validation, JMLR, 8: 985-1005, 2007.
>
> [R2] You, et al., Towards Accurate Model Selection in Deep Unsupervised Domain Adaptation, ICML, 2019.

---

> > ### Comment · Reviewer_Xqv1 · 2021-09-05
> > **Reply to Authors**
> >
> > Dear Authors,
> >
> > Thank you for your responses. I'm convinced that the experiments are sufficient. However, I agree with the other reviewers that the novelty of this work is limited. Hence, I will keep my original score.

---

> ### Author Response · Authors · 2021-08-10
> **Response to Reviewer Xqv1 Part(1/2)**
>
> We thank the reviewer for the comments and suggestions. We will revise the paper accordingly.
>
> **Q1**: Fair comparison with BA$^3$US and other methods.
>
> For fair comparison with BA$^3$US and other methods, we updated results on Office-31, Office-Home, ImageNet-Caltech, VisDA-2017, and DomainNet datasets, reported in Tables r3-1, r3-2, r3-3, r3-4, and r3-5, respectively. Since we used the spherical logistic regression (SLR) layer as the classifier, for completely fair comparison, we additionally report the results of the version of our AR with linear layer as the classifier (denoted as AR (w/ linear)). We also use "Cls+Ent" to denote the baseline approach that minimizes entropy loss on the target domain and source classification loss without using reweighting in Eq. (1). "Cls+Ent (w/ linear)" denotes the version of Cls+Ent with linear layer as the classifier.  In Tables r3-2, r3-3, r3-4, and r3-5, we observe that our AR (w/ linear) outperforms compared methods on Office-Home, ImageNet-Caltech, VisDA-2017, and DomainNet datasets. In Table r3-1, AR (w/ linear) also performs comparably to other compared methods on Office-31.
> We will add two rows for AR (w/ linear) and Cls+Ent (w/ linear) in Tables 2, 3, and 4 in our main paper, and replace the results of BAA with these of BA$^3$US. Some statements about results comparison will be also revised in Section 4.1 of the main paper.
>
> Table r3-1. Accuracy (\%) on Office-31 for partial domain adaptation.
>
> |Method|A$\rightarrow$D|A$\rightarrow$W|D$\rightarrow$A|D$\rightarrow$W|W$\rightarrow$A|W$\rightarrow$D|Avg|
> |:--|:--:|:--:|:--:|:--:|:--:|:--:|:--:|
> |ResNet-50 [15]|83.44|75.59|83.92|96.27|84.97|98.09|87.05|
> |DAN [23]|61.78|59.32|74.95|73.90|67.64|90.45|71.34|
> |DANN [9]|81.53|73.56|82.78|96.27|86.12|98.73|86.50|
> |IWAN [46]|90.45|89.15|95.62|99.32|94.26|99.36|94.69|
> |SAN [3]|94.27|93.90|94.15|99.32|88.73|99.36|94.96|
> |PADA [4]|82.17|86.54|92.69|99.32|95.41|__100.0__|92.69|
> |ETN [5]|95.03|94.52|96.21|__100.0__|94.64|__100.0__|96.73|
> |DRCN [21]|86.00|88.50|95.60|__100.0__|95.80|__100.0__|94.30|
> |RTNet$_{adv}$ [6]|96.20|97.60|92.30|__100.0__|95.40|__100.0__|97.20|
> |BA$^3$US [22]|__99.36__|__98.98__|94.82|__100.0__|94.99|98.73|__97.81__|
> |DPDAN [43]|96.27|96.82|__96.35__|__100.0__|95.62|__100.0__|97.51|
> |__Cls+Ent (w/ linear)__|90.45|87.80|94.68|__100.0__|94.36|98.09|94.23|
> |__AR (w/ linear) (ours)__|91.72|97.63|95.62|__100.0__|95.30|__100.0__|96.71|
> |__Cls+Ent__|80.89|87.12|94.05|94.58|93.95|99.36|91.66|
> |__AR (ours)__|96.82|93.54|95.51|__100.0__|__96.04__|99.67|96.93|
>
> Table r3-2. Accuracy (\%) on Office-Home for partial domain adaptation.
>
> |Method|Ar$\rightarrow$Cl|Ar$\rightarrow$Pr|Ar$\rightarrow$Rw|Cl$\rightarrow$Ar|Cl$\rightarrow$Pr|Cl$\rightarrow$Rw|Pr$\rightarrow$Ar|Pr$\rightarrow$Cl|Pr$\rightarrow$Rw|Rw$\rightarrow$Ar|Rw$\rightarrow$Cl|Rw$\rightarrow$Pr|Avg|
> |:--|:--:|:--:|:--:|:--:|:--:|:--:|:--:|:--:|:--:|:--:|:--:|:--:|:--:|
> |ResNet-50 [15]|46.33|67.51|75.87|59.14|59.94|62.73|58.22|41.79|74.88|67.40|48.18|74.17|61.35|
> |ADDA [39]|45.23|68.79|79.21|64.56|60.01|68.29|57.56|38.89|77.45|70.28|45.23|78.32|62.82|
> |CDAN+E [24]|47.52|65.91|75.65|57.07|54.12|63.42|59.60|44.30|72.39|66.02|49.91|72.80|60.73|
> |IWAN [46]|53.94|54.45|78.12|61.31|47.95|63.32|54.17|52.02|81.28|76.46|56.75|82.90|63.56|
> |SAN [3]|44.42|68.68|74.60|67.49|64.99|77.80|59.78|44.72|80.07|72.18|50.21|78.66|65.30|
> |PADA [4]|51.95|67.00|78.74|52.16|53.78|59.03|52.61|43.22|78.79|73.73|56.60|77.09|62.06|
> |ETN [5]|59.24|77.03|79.54|62.92|65.73|75.01|68.29|55.37|84.37|75.72|57.66|84.54|70.45|
> |DRCN [21]|54.00|76.40|83.00|62.10|64.50|71.00|70.80|49.80|80.50|77.50|59.10|79.90|69.00|
> |SAFN [42]|58.93|76.25|81.42|70.43|72.97|77.78|72.36|55.34|80.40|75.81|60.42|79.92|71.83|
> |RTNet$_{adv}$ [6]|63.20|80.10|80.70|66.70|69.30|77.20|71.60|53.90|84.60|77.40|57.90|85.50|72.30|
> |BA$^3$US [22]|60.62|83.16|88.39|71.75|72.79|83.40|75.45|61.59|86.53|79.25|62.80|86.05|75.98|
> |DPDAN [43]|59.40|-|79.04|-|-|-|-|-|81.79|76.77|58.67|82.18|-|
> |__Cls+Ent (w/ linear)__|54.03|73.61|83.27|69.51|67.56|77.75|69.51|53.73|83.38|74.56|59.34|82.41|70.72|
> |__AR (w/ linear) (ours)__|62.13|79.22|89.12|73.92|__75.57__|84.37|78.42|61.91|87.85|__82.19__|65.37|85.27|77.11|
> |__Cls+Ent__|61.61|78.21|86.20|73.19|71.76|79.62|75.11|59.76|86.31|79.16|61.67|83.59|74.68|
> |__AR (ours)__|__67.40__|__85.32__|__90.00__|__77.32__|70.59|__85.15__|__78.97__|__64.78__|__89.51__|80.44|__66.21__|__86.44__|__78.29__|
>
> Table r3-3. Accuracy (\%) on ImageNet-Caltech for partial domain adaptation.
>
> |Method|C$\rightarrow$I|I$\rightarrow$C|Avg|
> |:--|:--:|:--:|:--:|
> |ResNet-50 [15]|71.29|69.69|70.49|
> |DAN [23]|60.13|71.30|65.72|
> |DANN [9]|67.71|70.80|69.23|
> |IWAN [46]|73.33|78.06|75.70|
> |SAN [3]|75.26|77.75|76.51|
> |PADA [4]|70.48|75.03|72.76|
> |ETN [5]|74.93|83.23|79.08|
> |DRCN [21]|78.90|75.30|77.10|
> |BA$^3$US [22]|__83.35__|84.00|83.68|
> |__Cls+Ent (w/ linear)__|77.74|77.82|77.78|
> |__AR (w/ linear) (ours)__|81.78|85.83|83.81|
> |__Cls+Ent__|79.60|82.59|81.10|
> |__AR (ours)__|82.24|__87.12__|__84.69__|
>
> Table r3-4. Accuracy (\%) on VisDA-2017 for partial domain adaptation.
>
> |Method|R$\rightarrow$S|S$\rightarrow$R|Avg|
> |:--|:--:|:--:|:--:|
> |ResNet-50 [15]|64.28|45.26|54.77|
> |DAN [23]|68.35|47.60|57.98|
> |DANN [9]|73.84|51.01|62.43|
> |IWAN [46]|71.30|48.60|59.95|
> |SAN [3]|69.70|49.90|59.80|
> |PADA [4]|76.50|53.50|65.00|
> |DRCN [21]|73.20|58.20|65.70|
> |BA$^3$US [22]|67.56|69.86|68.71|
> |DPDAN [43]|-|65.26|-|
> |__Cls+Ent (w/ linear)__|69.00|82.32|75.66|
> |__AR (w/ linear) (ours)__|74.82|85.30|80.09|
> |__Cls+Ent__|66.63|84.72|75.68|
> |__AR (ours)__|__78.52__|__88.75__|__83.62__|
>
> Table r3-5. Accuracy (\%) on DomainNet for partial domain adaptation.
>
> |Method|C$\rightarrow$P|C$\rightarrow$R|C$\rightarrow$S|P$\rightarrow$C|P$\rightarrow$R|P$\rightarrow$S|R$\rightarrow$C|R$\rightarrow$P|R$\rightarrow$S|S$\rightarrow$C|S$\rightarrow$P|S$\rightarrow$R|Avg|
> |:--|:--:|:--:|:--:|:--:|:--:|:--:|:--:|:--:|:--:|:--:|:--:|:--:|:--:|
> |ResNet-50 [15]|41.21|60.01|42.13|54.52|70.80|48.32|63.10|58.63|50.26|45.43|39.30|49.75|51.96|
> |DANN [9]|27.83|36.64|29.91|31.79|41.98|36.58|47.64|46.81|40.85|25.82|29.54|32.72|35.68|
> |CDAN+E [24]|37.46|48.26|46.61|45.50|60.96|52.63|62.01|60.63|54.74|35.37|38.50|43.63|48.86|
> |SAN [3]|34.35|51.62|46.23|57.13|70.21|58.25|69.61|67.49|67.88|41.69|41.15|48.44|54.50|
> |PADA [4]|22.49|32.85|29.95|25.71|56.47|30.45|65.28|63.35|54.17|17.45|23.89|26.91|37.41|
> |BA$^3$US [22]|42.87|54.72|53.79|64.03|76.39|64.69|__79.99__|__74.31__|__74.02__|50.36|42.69|49.65|60.63|
> |__Cls+Ent (w/ linear)__|50.14|64.05|__59.81__|65.26|76.12|69.50|75.54|69.74|68.55|50.63|54.95|54.44|63.23|
> |__AR (w/ linear) (ours)__|__56.70__|__70.36__|58.56|65.63|74.80|__74.85__|75.22|71.17|69.08|__53.90__|__55.70__|__63.09__|__65.76__|
> |__Cls+Ent__|49.40|65.69|58.89|65.92|74.82|70.77|75.87|70.72|68.26|50.45|__55.70__|62.23|64.06|
> |__AR (ours)__|52.66|68.24|58.29|__66.78__|__77.53__|74.38|76.70|71.77|70.48|53.66|53.60|61.57|65.47|
>
> **Q2**: Clarifying the results of the Baseline.
>
> In Tables 2, 3, and 4 in the main paper, the "Baseline" denotes the approach that minimizes the source classification loss and the conditional entropy loss on the target domain. To avoid misunderstandings, we use "Cls+Ent" to replace "Baseline" in the main paper. In the updated results in Tables r3-1, r3-2, r3-3, r3-4, and r3-5, the Cls+Ent (w/ linear) and AR (w/ linear) are fair compared with prior methods. The Cls+Ent (w/ linear) outperforms the compared PDA methods mainly on VisDA-2017 and DomainNet datasets. This is well consistent with our finding in Section 2 that the reweighted distribution alignment in compared PDA methods may lead to negative domain transfer on these two datasets.

---

### Official Review · Reviewer_Hqei · 2021-07-16

**Rating:** 5
**Confidence:** 4

**Summary:**

This paper proposes a partial domain adaptation method, through an adversarial reweighting approach. Instead of learning a domain-invariant feature space, the proposed method reduces the domain shift through a source data re-weighting method.

**Limitations And Societal Impact:**

- It claims that previous PDA methods cannot work well on challenging benchmarks, like VisDA-2017 and DomainNet. It would be better to give more details on what aspects are these benchmarks more challenging, and compared to which benchmarks?

- The main cause of mistakenly assigning high weights to the source-only-class data for the previous methods is not very convincing. What are the key insights of the causes? It is good that Section 2 demonstrates the limitations of the previous methods through some experimental findings. However, it lacks more in-depth analysis, and thus the justifications are not convincing enough.

- It claims that previous PDA methods heuristically setting the weights of the source data, which is not precise. For example, the discriminator-based methods [5, 46] also automatically learn the source data weights through adversarial training rather than heuristically setting them.

- How/why does the proposed method can deal with the limitations of previous PDA methods? The proposed method assume that by re-weighting the source domain data, the domain distribution can be mitigated. However, this assumption is very strong. The domain shift between the source and target domains may not be able to be reduced by simply re-weighting the source domain data, especially for the so-called challenging benchmarks with large domain shift, which motivates the subsequent UDA methods that reduce the domain shift by learning a domain-invariant feature space.

- It argues that the proposed adversarial learning-based re-weighting method is different from previous kernel MMD-based methods and optimal transport model. However, to my understanding, they are all very commonly used criteria for aligning different distributions in domain adaptation, the only difference is the chosen metric/divergence for measuring the distribution shift. Thus, the novelty of the re-weighting method is limited.

- The comparisons to the previous method in the experiments part are not fair and complete. First, the author adopts different network architecture compared to previous PDA methods. For example, a state-of-the-art spherical network is adopted for the classifier C. Second, some comparison results are missing. For example, the results on ImageNet-Caltech, DomainNet, and VisDA-17 of some previous PDA methods that perform better than the proposed method on Office-31 are missing, e.g. RTNet[6] and DPDAN[43]. Thus, it is hard to judge the superior of the proposed method over these methods.

- The analysis of the experimental results is not sufficient, especially on why does the proposed method only perform comparably to other PDA methods on Office-31 dataset?



**Main Review:**

In general, the paper is well written and easy to follow. The proposed method has some novelty. It identifies some of the limitations of previous PDA methods and proposes a re-weighting model for re-weight the source data to achieve distribution alignment, rather than learning a domain-invariant feature space, to alliterate negative transfer in PDA. However, some of the statements are not very convincing, the novelty is limited, and the experiments are not fair and complete enough. The details can be found in the limitation part.

**Time Spent Reviewing:**

6

---

> ### Author Response · Authors · 2021-08-10
> **Response to Reviewer Hqei Part (2/2)**
>
>
> **Q8**: Fair and complete comparisons in experiments.
>
> Since we used the spherical logistic regression (SLR) layer as the classifier, for completely fair comparison, we also report the results of the version of our AR with linear layer as the classifier (denoted as AR (w/ linear)). We also use "Cls+Ent" to denote the baseline approach that minimizes entropy loss on the target domain and source classification loss without using reweighting in Eq. (1). "Cls+Ent (w/ linear)" denotes the version of Cls+Ent with linear layer as the classifier.  The updated results on Office-31, Office-Home, ImageNet-Caltech, VisDA-2017, and DomainNet datasets are reported in Tables r2-3, r2-4, r2-5, r2-6, and r2-7, respectively. We also reported the results of BA$^3$US on all five datasets. Since RTNet$_{adv}$ and DPDAN do not release their code, we are trying to produce their results on ImageNet-Caltech, VisDA-2017, and DomainNet datasets, which will be included in our paper if possible. In Tables  r2-4, r2-5, r2-6, and r2-7, we observe that our AR (w/ linear) outperforms compared methods on Office-Home, ImageNet-Caltech, VisDA-2017, and DomainNet datasets. In Table r2-3, AR (w/ linear) also performs comparably to other compared methods on Office-31.
> We will add two rows for AR (w/ linear) and Cls+Ent (w/ linear) in Tables 2, 3, and 4 in our main paper, and replace the results of BAA with these of BA$^3$US. Some statements about results comparison will be also revised in Section 4.1 of the main paper. The symbol "Baseline" will be replaced by "Cls+Ent".
>
> Table r2-3. Accuracy (\%) on Office-31 for partial domain adaptation.
>
> |Method|A$\rightarrow$D|A$\rightarrow$W|D$\rightarrow$A|D$\rightarrow$W|W$\rightarrow$A|W$\rightarrow$D|Avg|
> |:--|:--:|:--:|:--:|:--:|:--:|:--:|:--:|
> |ResNet-50 [15]|83.44|75.59|83.92|96.27|84.97|98.09|87.05|
> |DAN [23]|61.78|59.32|74.95|73.90|67.64|90.45|71.34|
> |DANN [9]|81.53|73.56|82.78|96.27|86.12|98.73|86.50|
> |IWAN [46]|90.45|89.15|95.62|99.32|94.26|99.36|94.69|
> |SAN [3]|94.27|93.90|94.15|99.32|88.73|99.36|94.96|
> |PADA [4]|82.17|86.54|92.69|99.32|95.41|__100.0__|92.69|
> |ETN [5]|95.03|94.52|96.21|__100.0__|94.64|__100.0__|96.73|
> |DRCN [21]|86.00|88.50|95.60|__100.0__|95.80|__100.0__|94.30|
> |RTNet$_{adv}$ [6]|96.20|97.60|92.30|__100.0__|95.40|__100.0__|97.20|
> |BA$^3$US [22]|__99.36__|__98.98__|94.82|__100.0__|94.99|98.73|__97.81__|
> |DPDAN [43]|96.27|96.82|__96.35__|__100.0__|95.62|__100.0__|97.51|
> |__Cls+Ent (w/ linear)__|90.45|87.80|94.68|__100.0__|94.36|98.09|94.23|
> |__AR (w/ linear) (ours)__|91.72|97.63|95.62|__100.0__|95.30|__100.0__|96.71|
> |__Cls+Ent__|80.89|87.12|94.05|94.58|93.95|99.36|91.66|
> |__AR (ours)__|96.82|93.54|95.51|__100.0__|__96.04__|99.67|96.93|
>
> Table r2-4. Accuracy (\%) on Office-Home for partial domain adaptation.
>
> |Method|Ar$\rightarrow$Cl|Ar$\rightarrow$Pr|Ar$\rightarrow$Rw|Cl$\rightarrow$Ar|Cl$\rightarrow$Pr|Cl$\rightarrow$Rw|Pr$\rightarrow$Ar|Pr$\rightarrow$Cl|Pr$\rightarrow$Rw|Rw$\rightarrow$Ar|Rw$\rightarrow$Cl|Rw$\rightarrow$Pr|Avg|
> |:--|:--:|:--:|:--:|:--:|:--:|:--:|:--:|:--:|:--:|:--:|:--:|:--:|:--:|
> |ResNet-50 [15]|46.33|67.51|75.87|59.14|59.94|62.73|58.22|41.79|74.88|67.40|48.18|74.17|61.35|
> |ADDA [39]|45.23|68.79|79.21|64.56|60.01|68.29|57.56|38.89|77.45|70.28|45.23|78.32|62.82|
> |CDAN+E [24]|47.52|65.91|75.65|57.07|54.12|63.42|59.60|44.30|72.39|66.02|49.91|72.80|60.73|
> |IWAN [46]|53.94|54.45|78.12|61.31|47.95|63.32|54.17|52.02|81.28|76.46|56.75|82.90|63.56|
> |SAN [3]|44.42|68.68|74.60|67.49|64.99|77.80|59.78|44.72|80.07|72.18|50.21|78.66|65.30|
> |PADA [4]|51.95|67.00|78.74|52.16|53.78|59.03|52.61|43.22|78.79|73.73|56.60|77.09|62.06|
> |ETN [5]|59.24|77.03|79.54|62.92|65.73|75.01|68.29|55.37|84.37|75.72|57.66|84.54|70.45|
> |DRCN [21]|54.00|76.40|83.00|62.10|64.50|71.00|70.80|49.80|80.50|77.50|59.10|79.90|69.00|
> |SAFN [42]|58.93|76.25|81.42|70.43|72.97|77.78|72.36|55.34|80.40|75.81|60.42|79.92|71.83|
> |RTNet$_{adv}$ [6]|63.20|80.10|80.70|66.70|69.30|77.20|71.60|53.90|84.60|77.40|57.90|85.50|72.30|
> |BA$^3$US [22]|60.62|83.16|88.39|71.75|72.79|83.40|75.45|61.59|86.53|79.25|62.80|86.05|75.98|
> |DPDAN [43]|59.40|-|79.04|-|-|-|-|-|81.79|76.77|58.67|82.18|-|
> |__Cls+Ent (w/ linear)__|54.03|73.61|83.27|69.51|67.56|77.75|69.51|53.73|83.38|74.56|59.34|82.41|70.72|
> |__AR (w/ linear) (ours)__|62.13|79.22|89.12|73.92|__75.57__|84.37|78.42|61.91|87.85|__82.19__|65.37|85.27|77.11|
> |__Cls+Ent__|61.61|78.21|86.20|73.19|71.76|79.62|75.11|59.76|86.31|79.16|61.67|83.59|74.68|
> |__AR (ours)__|__67.40__|__85.32__|__90.00__|__77.32__|70.59|__85.15__|__78.97__|__64.78__|__89.51__|80.44|__66.21__|__86.44__|__78.29__|
>
>
> Table r2-5. Accuracy (\%) on ImageNet-Caltech for partial domain adaptation.
>
> |Method|C$\rightarrow$I|I$\rightarrow$C|Avg|
> |:--|:--:|:--:|:--:|
> |ResNet-50 [15]|71.29|69.69|70.49|
> |DAN [23]|60.13|71.30|65.72|
> |DANN [9]|67.71|70.80|69.23|
> |IWAN [46]|73.33|78.06|75.70|
> |SAN [3]|75.26|77.75|76.51|
> |PADA [4]|70.48|75.03|72.76|
> |ETN [5]|74.93|83.23|79.08|
> |DRCN [21]|78.90|75.30|77.10|
> |BA$^3$US [22]|__83.35__|84.00|83.68|
> |__Cls+Ent (w/ linear)__|77.74|77.82|77.78|
> |__AR (w/ linear) (ours)__|81.78|85.83|83.81|
> |__Cls+Ent__|79.60|82.59|81.10|
> |__AR (ours)__|82.24|__87.12__|__84.69__|
>
> Table r2-6. Accuracy (\%) on VisDA-2017 for partial domain adaptation.
>
> |Method|R$\rightarrow$S|S$\rightarrow$R|Avg|
> |:--|:--:|:--:|:--:|
> |ResNet-50 [15]|64.28|45.26|54.77|
> |DAN [23]|68.35|47.60|57.98|
> |DANN [9]|73.84|51.01|62.43|
> |IWAN [46]|71.30|48.60|59.95|
> |SAN [3]|69.70|49.90|59.80|
> |PADA [4]|76.50|53.50|65.00|
> |DRCN [21]|73.20|58.20|65.70|
> |BA$^3$US [22]|67.56|69.86|68.71|
> |DPDAN [43]|-|65.26|-|
> |__Cls+Ent (w/ linear)__|69.00|82.32|75.66|
> |__AR (w/ linear) (ours)__|74.82|85.30|80.09|
> |__Cls+Ent__|66.63|84.72|75.68|
> |__AR (ours)__|__78.52__|__88.75__|__83.62__|
>
> Table r2-7. Accuracy (\%) on DomainNet for partial domain adaptation.
>
> |Method|C$\rightarrow$P|C$\rightarrow$R|C$\rightarrow$S|P$\rightarrow$C|P$\rightarrow$R|P$\rightarrow$S|R$\rightarrow$C|R$\rightarrow$P|R$\rightarrow$S|S$\rightarrow$C|S$\rightarrow$P|S$\rightarrow$R|Avg|
> |:--|:--:|:--:|:--:|:--:|:--:|:--:|:--:|:--:|:--:|:--:|:--:|:--:|:--:|
> |ResNet-50 [15]|41.21|60.01|42.13|54.52|70.80|48.32|63.10|58.63|50.26|45.43|39.30|49.75|51.96|
> |DANN [9]|27.83|36.64|29.91|31.79|41.98|36.58|47.64|46.81|40.85|25.82|29.54|32.72|35.68|
> |CDAN+E [24]|37.46|48.26|46.61|45.50|60.96|52.63|62.01|60.63|54.74|35.37|38.50|43.63|48.86|
> |SAN [3]|34.35|51.62|46.23|57.13|70.21|58.25|69.61|67.49|67.88|41.69|41.15|48.44|54.50|
> |PADA [4]|22.49|32.85|29.95|25.71|56.47|30.45|65.28|63.35|54.17|17.45|23.89|26.91|37.41|
> |BA$^3$US [22]|42.87|54.72|53.79|64.03|76.39|64.69|__79.99__|__74.31__|__74.02__|50.36|42.69|49.65|60.63|
> |__Cls+Ent (w/ linear)__|50.14|64.05|__59.81__|65.26|76.12|69.50|75.54|69.74|68.55|50.63|54.95|54.44|63.23|
> |__AR (w/ linear) (ours)__|__56.70__|__70.36__|58.56|65.63|74.80|__74.85__|75.22|71.17|69.08|__53.90__|__55.70__|__63.09__|__65.76__|
> |__Cls+Ent__|49.40|65.69|58.89|65.92|74.82|70.77|75.87|70.72|68.26|50.45|__55.70__|62.23|64.06|
> |__AR (ours)__|52.66|68.24|58.29|__66.78__|__77.53__|74.38|76.70|71.77|70.48|53.66|53.60|61.57|65.47|
>
> **Q9**: Analysis on why AR only performs comparably to other PDA methods on Office-31 dataset.
>
> In Table r2-3, we can see that the accuracies of RTNet, BA$^3$US, and DPDAN on office-31 are higher than 97\%. This indicates that the prediction of the classifier on the target domain may be reliable and the source weights based on the classifier may contain less noise. In such a case, positive domain transfer may be achieved by feature adaptation. Note that although RTNet does not explicitly reweight data using the classifier outputs, the classifier outputs on the target domain are utilized as the component of state in a reinforcement learning framework to select the source data for feature adaptation. Hence, better classifier prediction may be beneficial for performance improvement. On the other datasets (e.g., Office-Home, ImageNet-Caltech, VisDA-2017, and DomainNet), the average accuracies of compared methods are all lower than 85\%, largely lower than the accuracies on Office-31. Then, the prediction of the classifier on the target domain may not be reliable as Office-31, and our AR outperforms the compared methods on these datasets. This may be because our AR could be more robust to weight noise than other methods. We will include this analysis in Section 4.1 of the main paper.
>
> **Q10**: Measuring the hardness of a dataset for partial domain adaptation using weight noise (refer to lines 144-145 in Section 2) level.
>
> We propose to use weight noise level to measure the hardness of a dataset for partial domain adaptation.  The weight noise level of a dataset is defined as follows. We first train a model using the baseline approach that minimizes the source classification loss and entropy loss on target domain. The weight noise level of a patial domain adaptation task is the average predicted probabilty of the target domain samples being misclassifed into the source-only classes, using the trained model. The average weight noise level of all tasks in the dataset is taken as the weight noise level.  We report the noise levels of five PDA benchmark datasets in Table r2-8.
>
> Table r2-8. Noise level of five benchmark datasets.
>
> ||Office-31|Office-Home|ImageNet-Caltech|VisDA-2017|DomainNet|
> |:--:|:--:|:--:|:--:|:--:|:--:|
> |Noise level| 0.15|0.21|0.20|0.30|0.42|
>
> We find that VisDA-2017 and DomainNet datasets introduce larger weight noise levels, i.e., more target domain samples are misclassified to the source-only classes. This indicates that VisDA-2017 and DomainNet are more challenging than the other datasets for PDA. We will include the weight noise levels as a barplot in Section 2 in the main paper.

---

> > ### Comment · Reviewer_Hqei · 2021-09-05
> > **Reply to the rebuttal**
> >
> > Thanks for the detailed results. However, novelty is still my main concern. So I will keep my initial score.

---

> ### Author Response · Authors · 2021-08-10
> **Response to Reviewer Hqei Part (1/2)**
>
> We thank the reviewer for the comments and suggestions. We will revise the paper accordingly.
>
>
> **Q1**: About the "challenging" benchmarks.
>
> VisDA-2017 [31] contains 280,157 images, much larger than the scale (1,410) of Office-31. VisDA-2017 focuses on the simulation-to-reality shift [31]. The domain gaps within VisDA-2017 may be larger than Office-31 with real-to-real shift.  DomainNet [30] contains 569,010 images from 6 domains sharing 354 classes. We follow [34] to use its subset with 126 classes from 4 domains. The scale, number of classes, number of domains, and gaps between domains of DomainNet are larger than these of Office-31.
> We find that on VisDA and DomainNet datasets, additionally utilizing the reweighted distribution alignment losses of PADA, BAA, and MMD degrades the performance of the baseline method, which uses only source classification loss and entropy loss on target domain.  We will revise the statements in lines 40-41. We also propose to use weight noise level to measure the hardness of a dataset for partial domain adaptation (please refer to the response to Q10).
>
>
> **Q2**: Cause of assigning non-zero weights to the source-only-class data.
>
> In the PDA methods of SAN [3], PADA [4], BA$^3$US [22], the weight of source domain data is defined using the average predicted classification probability on the target domain data. When the domain gap between source and target domains is large, the prediction by the source classifier on the target domain may be uncertain. Then the predicted probability of classifying the target data as source-only classes may be non-zero or even possibly significantly larger than zero. Hence, reweighting source classes based on the outputs of the classifier may assign non-zero weights to the source-only classes.
>
>
> **Q3**: Analysis of experimental findings in Section 2.
>
> In Table 1 of the main paper, "w/o Alignment" denotes the approach that minimizes the source classification loss and entropy loss on the target domain. "PADA"/"BAA"/"MMD" denote the approaches that additionally minimize the reweighted distribution alignment losses of PADA/BAA/MMD, compared with "w/o Alignment".
> We find that "PADA"/"BAA"/"MMD" degrade the performance of "w/o Alignment" on VisDA-2017 and DomainNet datasets. In Fig. 1 of the main paper, we conduct a simulation experiment to testify the robustness of the reweighted distribution alignment losses to weight noise. From Fig. 1, we can see that when the noise level is near zero, all the alignment losses outperform "w/o Alignment", indicating that the alignment losses may lead to positive domain transfer in this case. However, as the noise level increases, the performance of the alignment losses decreases rapidly and even becomes significantly worse than the "w/o Alignment". Specifically, in task S$\rightarrow$R, when the noise level is 0.1, the alignment losses begin to degrade the result of the "w/o Alignment". Similarly, in task C$\rightarrow$P, the alignment losses begin to degrade the performance of the "w/o Alignment" at noise level 0.3. While the "real" noise level (the sum of weights over source-only classes) is larger than 0.3 (resp. 0.4) in the two tasks, respectively. Therefore, the negative domain transfer of the reweighted distribution alignment losses may be because of the “noise” in weights. We will revise the analysis in Section 2 (lines 143-148) according to this discussion.
>
>
> **Q4**: Correcting the statement "heuristically setting the weights of the source data in previous PDA methods".
>
> Thanks. We will correct the statement in lines 55-61. The revised statement could be "The current PDA methods [3, 4, 5, 21, 22, 43, 46] design/learn source data weights based on the classifier [3, 4, 21, 22, 43] or discriminator [5, 46]. They then train the feature extractor using a reweighted distribution alignment loss defined on the target and reweighted source data. Different from them, firstly, we learn the weights of source data in our proposed adversarial reweighting model to decrease the weight of source-only-class data.  Secondly, we reduce the domain gap by reweighting the source domain data, instead of directly optimizing the feature extractor to match feature distributions".
>
>
> **Q5**: Verification of the robustness of our AR to weight noise (answering why AR may be able to deal with the limitation of some of previous methods).
>
> To verify the robustness of AR w.r.t. weight noise, we conduct simulation experiments for our method under different noise levels. Specifically, we first obtain the source data weights $w_i, i=1,2,..,n_s$, through our adversarial reweighting model. For the noise level $p$, we simulate the weight of the $i$-th sample that belongs to source-only classes by $p*w_i/\sum_{j=1}^{n_s}I(y^s_j\in \mathcal{Y}^s\backslash\mathcal{Y}^t)w_j$, for $i$ such that $y^s_i\in \mathcal{Y}^s\backslash\mathcal{Y}^t$ (i.e., the $i$-th source sample that belongs to source-only classes is assigned noisy weight in proportional to the weight learned by our adversarial reweighting model). Similarly, the $i$-th sample that belongs to source-shared classes is assigned noisy weight by $(1-p)*w_i/\sum_{j=1}^{n_s}I(y^s_j\in\mathcal{Y}^t)w_j$, for $i$ such that $y^s_i\in \mathcal{Y}^t$. The results in the tasks S$\rightarrow$R on VisDA-2017 and C$\rightarrow$P on DomainNet are reported in Tables r2-1 and r2-2.
>
> Table r2-1. Results for AR with varying noise levels in the task S$\rightarrow$R on VisDA-2017.
>
> |Noise level|0.0|0.1|0.2|0.3|0.4|
> |:--|:--:|:--:|:--:|:--:|:--:|
> |Cls+Ent|84.72|84.72|84.72|84.72|84.72|
> |AR|89.76|88.01|88.74|86.02 |85.83 |
>
> Table r2-2. Results for AR with varying noise levels in the task C$\rightarrow$P on DomainNet.
>
> |Noise level|0.0|0.1|0.2|0.3|0.4|
> |:--|:--:|:--:|:--:|:--:|:--:|
> |Cls+Ent|49.40|49.40|49.40|49.40|49.40|
> |AR|56.86|55.42|53.97|52.17|51.45|
>
> Cls+Ent denotes the baseline of AR that minimizes entropy loss on the target domain and source classification loss (without reweighting).
> Since the Cls+Ent does not use reweighting, its results are the same under different noise levels. We observe that our AR consistently outperforms Cls+Ent when the noise level is not higher than 0.4. We will include these results as figures in our main paper.
>
>
> **Q6**: Why AR can work with larger domain shift.
>
> In our methods, besides the source classification loss, we also minimize the entropy of predicted classification probability on the target domain (see Eq. (1)). As stated in line 175, the entropy loss is used only to update the feature extractor, instead of both classifier and feature extractor. Our classifier is the spherical logistic regression layer that outputs the cosine similarity of the target features and source prototypes [13]. With lower entropy, the learned target features need to be close to the source features so that the source classifier can recognize them more surely. Hence, our adversarial reweighting and entropy minimization complement each other to reduce the gap between source and target distributions. As an example, we visualize the learned features by the Cls+Ent and our AR in Fig. 4(c) and 4(d) in the main paper, respectively. We observe in Fig. 4(c) that the Cls+Ent (that minimizes the entropy without using reweighting) aligns target features with features of all source classes, which is unexpected in PDA. In Fig. 4(d), our AR (that utilizes both entropy minimization and adversarial reweighting) aligns target features with features of partial source classes. We will include this discussion at the end of Section 3.2 of our main paper, and revise the analysis between lines 318-324 in Section 4.
>
>
> **Q7**: Novelty of our method.
>
> We propose the novel adversarial reweighting, a simple but effective approach that is quite aligned with the PDA task. We learn to reweight the importance of source data for reducing the domain shift, measured by the Wasserstein distance. It is non-trivial to achieve our goal, due to the computational issue of the Wasserstein distance. We use the approximation of its dual form, deducing the adversarial reweighting model. We also propose an iterative training algorithm that alternately updates the network parameters and learns the source weights by adversarially training the discriminator and solving cone programming.

---

### Official Review · Reviewer_jtrd · 2021-07-17

**Rating:** 5
**Confidence:** 3

**Summary:**

This paper presents a novel approach for partial domain adaptation (PDA) based on the adversarial reweighting of source samples. The motivation of this paper is quite clear that the source private sample may cause negative transfer in such a partial domain adaptation setting. Reweighting is a common practice to address such challenges. In this paper, the authors propose an improved version that the sample weights can be updated with the help of adversarial learning based on a discriminator. It is a smart design that shows promising results.

**Limitations And Societal Impact:**

Yes. The authors provide adequate discussion of limitations and potential negative societal impact.

**Main Review:**

Strengths:
+ As mentioned above, the motivation of this paper is quite clear and convincing.

+ The proposed adversarial re-weighting technique is simple but seems effective.

+ The whole paper is organized logically, and the discussion of related works is comprehensive enough to cover most of the literature in this field.


Weaknesses:
- Although the motivation is clear, the whole paper is built upon the well-explored techniques such as sample re-weighting and adversarial training, making me feel like a combination. I am doubtful about the originality of this paper for the standard of NeurIPS.

- The proposed method seems too simple, and there is no significant improvement (comparing Baseline and AR) on some datasets like DomainNet.

Question:
1. I doubt the fair comparison between the proposed method and other baseline methods. It seems the naïve version (Baseline) outperforms other PDA methods by a large margin.

In all, I recommend the score 5 due to the weaknesses and questions. I will consider upgrading the score if the authors provide a strong rebuttal.


**Time Spent Reviewing:**

4

---

> ### Author Response · Authors · 2021-08-10
> **Response to Reviewer jtrd**
>
> We thank the reviewer for the comments.
>
> **Q1**: Our contributions.
>
> 1) We experimentally find that when the source data weights contain "noise", some state-of-the-art reweighted distribution alignment losses (e.g., the reweighted adversarial training losses in PADA [4] and BA$^3$US [22], and the reweighted MMD loss [21]) for PDA are not robust to the weight noise. On VisDA and DomainNet datasets with larger diversity and domain gaps, additionally utilizing the reweighted distribution alignment losses of PADA, BAA, and  MMD degrades the performance of the baseline method, which uses only source classification loss and entropy loss on the target domain.
>
> 2) We further propose a novel adversarial reweighting approach for PDA without using the reweighted distribution alignment losses. We learn to reweight the importance of source domain data for reducing the domain shift, measured by the Wasserstein distance. We then train the transferable model for target domain by the reweighted source classification loss that is shown to be more robust to weight noise (please refer to Q5 in the response to reviewer Hqei). Due to the computational issue of the Wasserstein distance, we use the approximation of its dual form, deducing the adversarial reweighting model. We also propose an iterative training algorithm that alternately updates the network parameters and learns the source weights by adversarially training the discriminator and solving cone programming. Extensive experiments on five benchmarks confirmed the effectiveness of the proposed approach.
>
> **Q2**: Fair comparison with prior methods.
>
> Since we used the spherical logistic regression (SLR) layer as the classifier, for completely fair comparison, we also report the results of the version of our AR with linear layer as the classifier (denoted as AR (w/ linear)). We also use "Cls+Ent" to denote the baseline approach that minimizes entropy loss on the target domain and source classification loss without using reweighting in Eq. (1). "Cls+Ent (w/ linear)" denotes the version of Cls+Ent with linear layer as the classifier.
> The updated results on Office-31, Office-Home, ImageNet-Caltech, VisDA-2017, and DomainNet datasets are reported in Tables r1-1, r1-2, r1-3, r1-4, and r1-5, respectively. In these tables, we also compare AR with BA$^3$US rather than BAA. In Tables r1-2, r1-3, r1-4, and r1-5, we observe that our AR (w/ linear) outperforms compared methods on Office-Home, ImageNet-Caltech, VisDA-2017, and DomainNet datasets. In Table r1-1, AR (w/ linear) also performs comparably to other compared methods on Office-31.
> We will add two rows for AR (w/ linear) and Cls+Ent (w/ linear) in Tables 2, 3, and 4 in the main paper, and replace the results of BAA with these of BA$^3$US. The symbol "Baseline" will be replaced by "Cls+Ent". Some statements about results comparison will be also revised in Section 4.1 of the main paper.
>
> Table r1-1. Accuracy (\%) on Office-31 for partial domain adaptation.
>
> |Method|A$\rightarrow$D|A$\rightarrow$W|D$\rightarrow$A|D$\rightarrow$W|W$\rightarrow$A|W$\rightarrow$D|Avg|
> |:--|:--:|:--:|:--:|:--:|:--:|:--:|:--:|
> |ResNet-50 [15]|83.44|75.59|83.92|96.27|84.97|98.09|87.05|
> |DAN [23]|61.78|59.32|74.95|73.90|67.64|90.45|71.34|
> |DANN [9]|81.53|73.56|82.78|96.27|86.12|98.73|86.50|
> |IWAN [46]|90.45|89.15|95.62|99.32|94.26|99.36|94.69|
> |SAN [3]|94.27|93.90|94.15|99.32|88.73|99.36|94.96|
> |PADA [4]|82.17|86.54|92.69|99.32|95.41|__100.0__|92.69|
> |ETN [5]|95.03|94.52|96.21|__100.0__|94.64|__100.0__|96.73|
> |DRCN [21]|86.00|88.50|95.60|__100.0__|95.80|__100.0__|94.30|
> |RTNet$_{adv}$ [6]|96.20|97.60|92.30|__100.0__|95.40|__100.0__|97.20|
> |BA$^3$US [22]|__99.36__|__98.98__|94.82|__100.0__|94.99|98.73|__97.81__|
> |DPDAN [43]|96.27|96.82|__96.35__|__100.0__|95.62|__100.0__|97.51|
> |__Cls+Ent (w/ linear)__|90.45|87.80|94.68|__100.0__|94.36|98.09|94.23|
> |__AR (w/ linear) (ours)__|91.72|97.63|95.62|__100.0__|95.30|__100.0__|96.71|
> |__Cls+Ent__|80.89|87.12|94.05|94.58|93.95|99.36|91.66|
> |__AR (ours)__|96.82|93.54|95.51|__100.0__|__96.04__|99.67|96.93|
>
> Table r1-2. Accuracy (\%) on Office-Home for partial domain adaptation.
>
> |Method|Ar$\rightarrow$Cl|Ar$\rightarrow$Pr|Ar$\rightarrow$Rw|Cl$\rightarrow$Ar|Cl$\rightarrow$Pr|Cl$\rightarrow$Rw|Pr$\rightarrow$Ar|Pr$\rightarrow$Cl|Pr$\rightarrow$Rw|Rw$\rightarrow$Ar|Rw$\rightarrow$Cl|Rw$\rightarrow$Pr|Avg|
> |:--|:--:|:--:|:--:|:--:|:--:|:--:|:--:|:--:|:--:|:--:|:--:|:--:|:--:|
> |ResNet-50 [15]|46.33|67.51|75.87|59.14|59.94|62.73|58.22|41.79|74.88|67.40|48.18|74.17|61.35|
> |ADDA [39]|45.23|68.79|79.21|64.56|60.01|68.29|57.56|38.89|77.45|70.28|45.23|78.32|62.82|
> |CDAN+E [24]|47.52|65.91|75.65|57.07|54.12|63.42|59.60|44.30|72.39|66.02|49.91|72.80|60.73|
> |IWAN [46]|53.94|54.45|78.12|61.31|47.95|63.32|54.17|52.02|81.28|76.46|56.75|82.90|63.56|
> |SAN [3]|44.42|68.68|74.60|67.49|64.99|77.80|59.78|44.72|80.07|72.18|50.21|78.66|65.30|
> |PADA [4]|51.95|67.00|78.74|52.16|53.78|59.03|52.61|43.22|78.79|73.73|56.60|77.09|62.06|
> |ETN [5]|59.24|77.03|79.54|62.92|65.73|75.01|68.29|55.37|84.37|75.72|57.66|84.54|70.45|
> |DRCN [21]|54.00|76.40|83.00|62.10|64.50|71.00|70.80|49.80|80.50|77.50|59.10|79.90|69.00|
> |SAFN [42]|58.93|76.25|81.42|70.43|72.97|77.78|72.36|55.34|80.40|75.81|60.42|79.92|71.83|
> |RTNet$_{adv}$ [6]|63.20|80.10|80.70|66.70|69.30|77.20|71.60|53.90|84.60|77.40|57.90|85.50|72.30|
> |BA$^3$US [22]|60.62|83.16|88.39|71.75|72.79|83.40|75.45|61.59|86.53|79.25|62.80|86.05|75.98|
> |DPDAN [43]|59.40|-|79.04|-|-|-|-|-|81.79|76.77|58.67|82.18|-|
> |__Cls+Ent (w/ linear)__|54.03|73.61|83.27|69.51|67.56|77.75|69.51|53.73|83.38|74.56|59.34|82.41|70.72|
> |__AR (w/ linear) (ours)__|62.13|79.22|89.12|73.92|__75.57__|84.37|78.42|61.91|87.85|__82.19__|65.37|85.27|77.11|
> |__Cls+Ent__|61.61|78.21|86.20|73.19|71.76|79.62|75.11|59.76|86.31|79.16|61.67|83.59|74.68|
> |__AR (ours)__|__67.40__|__85.32__|__90.00__|__77.32__|70.59|__85.15__|__78.97__|__64.78__|__89.51__|80.44|__66.21__|__86.44__|__78.29__|
>
>
> Table r1-3. Accuracy (\%) on ImageNet-Caltech for partial domain adaptation.
>
> |Method|C$\rightarrow$I|I$\rightarrow$C|Avg|
> |:--|:--:|:--:|:--:|
> |ResNet-50 [15]|71.29|69.69|70.49|
> |DAN [23]|60.13|71.30|65.72|
> |DANN [9]|67.71|70.80|69.23|
> |IWAN [46]|73.33|78.06|75.70|
> |SAN [3]|75.26|77.75|76.51|
> |PADA [4]|70.48|75.03|72.76|
> |ETN [5]|74.93|83.23|79.08|
> |DRCN [21]|78.90|75.30|77.10|
> |BA$^3$US [22]|__83.35__|84.00|83.68|
> |__Cls+Ent (w/ linear)__|77.74|77.82|77.78|
> |__AR (w/ linear) (ours)__|81.78|85.83|83.81|
> |__Cls+Ent__|79.60|82.59|81.10|
> |__AR (ours)__|82.24|__87.12__|__84.69__|
>
> Table r1-4. Accuracy (\%) on VisDA-2017 for partial domain adaptation.
>
> |Method|R$\rightarrow$S|S$\rightarrow$R|Avg|
> |:--|:--:|:--:|:--:|
> |ResNet-50 [15]|64.28|45.26|54.77|
> |DAN [23]|68.35|47.60|57.98|
> |DANN [9]|73.84|51.01|62.43|
> |IWAN [46]|71.30|48.60|59.95|
> |SAN [3]|69.70|49.90|59.80|
> |PADA [4]|76.50|53.50|65.00|
> |DRCN [21]|73.20|58.20|65.70|
> |BA$^3$US [22]|67.56|69.86|68.71|
> |DPDAN [43]|-|65.26|-|
> |__Cls+Ent (w/ linear)__|69.00|82.32|75.66|
> |__AR (w/ linear) (ours)__|74.82|85.30|80.09|
> |__Cls+Ent__|66.63|84.72|75.68|
> |__AR (ours)__|__78.52__|__88.75__|__83.62__|
>
> Table r1-5. Accuracy (\%) on DomainNet for partial domain adaptation.
>
> |Method|C$\rightarrow$P|C$\rightarrow$R|C$\rightarrow$S|P$\rightarrow$C|P$\rightarrow$R|P$\rightarrow$S|R$\rightarrow$C|R$\rightarrow$P|R$\rightarrow$S|S$\rightarrow$C|S$\rightarrow$P|S$\rightarrow$R|Avg|
> |:--|:--:|:--:|:--:|:--:|:--:|:--:|:--:|:--:|:--:|:--:|:--:|:--:|:--:|
> |ResNet-50 [15]|41.21|60.01|42.13|54.52|70.80|48.32|63.10|58.63|50.26|45.43|39.30|49.75|51.96|
> |DANN [9]|27.83|36.64|29.91|31.79|41.98|36.58|47.64|46.81|40.85|25.82|29.54|32.72|35.68|
> |CDAN+E [24]|37.46|48.26|46.61|45.50|60.96|52.63|62.01|60.63|54.74|35.37|38.50|43.63|48.86|
> |SAN [3]|34.35|51.62|46.23|57.13|70.21|58.25|69.61|67.49|67.88|41.69|41.15|48.44|54.50|
> |PADA [4]|22.49|32.85|29.95|25.71|56.47|30.45|65.28|63.35|54.17|17.45|23.89|26.91|37.41|
> |BA$^3$US [22]|42.87|54.72|53.79|64.03|76.39|64.69|__79.99__|__74.31__|__74.02__|50.36|42.69|49.65|60.63|
> |__Cls+Ent (w/ linear)__|50.14|64.05|__59.81__|65.26|76.12|69.50|75.54|69.74|68.55|50.63|54.95|54.44|63.23|
> |__AR (w/ linear) (ours)__|__56.70__|__70.36__|58.56|65.63|74.80|__74.85__|75.22|71.17|69.08|__53.90__|__55.70__|__63.09__|__65.76__|
> |__Cls+Ent__|49.40|65.69|58.89|65.92|74.82|70.77|75.87|70.72|68.26|50.45|__55.70__|62.23|64.06|
> |__AR (ours)__|52.66|68.24|58.29|__66.78__|__77.53__|74.38|76.70|71.77|70.48|53.66|53.60|61.57|65.47|
>
> **Q3**: Clarifying the results of "Baseline".
>
> The "Baseline" in Tables 2, 3, and 4 in the main paper denotes the approach that minimizes the source classification loss and the conditional entropy loss on the target domain. As mentioned in the response to Q2, we will use "Cls+Ent" to replace it. The reason that may lead to unfair comparison is the use of the spherical logistic regression layer as the classifier. In the updated results in Tables r1-1, r1-2, r1-3, r1-4, and r1-5, the Cls+Ent (w/ linear) and AR (w/ linear) are fair compared with prior methods. The Cls+Ent (w/ linear) outperforms the compared PDA methods mainly on VisDA-2017 and DomainNet datasets. This is well consistent with our finding in Section 2 that the reweighted distribution alignment losses in compared PDA methods may lead to negative domain transfer on these two datasets.
>
> **Q4**: Effectiveness of AR on DomainNet dataset.
>
> In Table r1-5, on DomainNet dataset, our AR (resp. AR (w/ linear)) outperforms the Cls+Ent (resp. Cls+Ent (w/ linear)) by 1.41\% (resp. 2.53\%). We find our AR  and AR (w/ linear) are the only two methods outperforming the baseline method of Cls+Ent (w/ linear), among all the compared methods.

---

> > ### Comment · Reviewer_jtrd · 2021-08-31
> > **Reply to Author**
> >
> > Dear Authors,
> >
> > Thanks for your detailed rebuttal. I am satisfied for the results on latest datasets. The major concern on novelty is still not addressed. The whole framework utilized the well explored techniques with straight forward combinations. I believe such technical novelty is slightly below the standard of NeurIPS and will keep my initial score.

---

### Decision · Program_Chairs · 2021-09-28

**Decision:**

Accept (Poster)

**Comment:**

This paper points out that the existing partial domain adaptation method has negative transfer when some source-only data are mistakenly assigned high weights. The authors thus propose an adversarial reweighting approach to minimize the distance between reweighted source and target domains. The proposed method outperforms existing methods on several domain adaptation datasets.

However, there are several concerns. 1) The motivation of the paper is not strong enough. The claims in the paper are not convincing. For example, the paper claims that existing PDA methods do not work well on the challenging datasets, such as visDA2017, but it does not explain why such datasets are challenging. Also, the analysis of the drawbacks of existing methods is not thorough enough. For example, the discriminator-based methods [5,46] also automatically learn the source domain weights using adversarial learning rather than heuristically setting them. 2) The proposed adversarial learning method differs from existing methods only in distribution divergence measures. The novelty of the proposed method is limited. 3) The performance of the proposed method does not clearly outperform existing methods by a large margin. Also, the comparison might not be fair because the authors aopt a different network architecture compared to previous PDA methods. The authors have added a long list of new results in the rebuttal, which means the manuscript needs significant modifications of the experimental section.

Overall, the paper will have some practical values if the experimental results are further strengthened. However, given the above concerns, especially those on the novelty of the paper, we think the paper does not meet the acceptance requirements of NeurIPS.


**Consistency Experiment:**

NeurIPS has a long history of experimentation. In 2014, NeurIPS ran an experiment in which 10% of submissions were reviewed by two independent committees to quantify the randomness in the review process. This year, we repeated a variant of this experiment to see how the quality of the review process has changed over time.  This paper was part of the experiment and was therefore assigned to two committees (consisting of reviewers, an Area Chair, and a Senior Area Chair) that reached independent decisions.  If both committees made the same recommendation, this recommendation was followed. If a single committee recommended acceptance, the paper was accepted (with the exception of a few cases in which the other committee identified what we considered a fatal flaw, e.g., an error in a key result).

This copy’s committee reached the following decision: **Reject**

The other committee assigned to the paper recommended **Accept (Poster)**.  You can find the other set of reviews, along with any follow up discussion with the authors here:
https://openreview.net/forum?id=f5liPryFRoA